# Classification of Explainable Artificial Intelligence Methods through Their Output Formats

Giulia Vilone *,† and Luca Longo †

Applied Intelligence Research Centre, Technological University Dublin, D08 X622 Dublin, Ireland; luca.longo@tudublin.ie

* Correspondence: giulia.vilone@tudublin.ie
† These authors contributed equally to this work.

**Abstract:** Machine and deep learning have proven their utility to generate data-driven models with high accuracy and precision. However, their non-linear, complex structures are often difficult to interpret. Consequently, many scholars have developed a plethora of methods to explain their functioning and the logic of their inferences. This systematic review aimed to organise these methods into a hierarchical classification system that builds upon and extends existing taxonomies by adding a significant dimension—the output formats. The reviewed scientific papers were retrieved by conducting an initial search on Google Scholar with the keywords "explainable artificial intelligence"; "explainable machine learning"; and "interpretable machine learning". A subsequent iterative search was carried out by checking the bibliography of these articles. The addition of the dimension of the explanation format makes the proposed classification system a practical tool for scholars, supporting them to select the most suitable type of explanation format for the problem at hand. Given the wide variety of challenges faced by researchers, the existing XAI methods provide several solutions to meet the requirements that differ considerably between the users, problems and application fields of artificial intelligence (AI). The task of identifying the most appropriate explanation can be daunting, thus the need for a classification system that helps with the selection of methods. This work concludes by critically identifying the limitations of the formats of explanations and by providing recommendations and possible future research directions on how to build a more generally applicable XAI method. Future work should be flexible enough to meet the many requirements posed by the widespread use of AI in several fields, and the new regulations.

**Keywords:** explainable artificial intelligence; method classification; systematic literature review

## 1. Introduction

In recent years, many scholars have devoted their efforts to the search for novel methods that are capable of exposing and explaining the logic followed by data-driven machine-learned models, thus creating a new subfield of artificial intelligence (AI) known as explainable artificial intelligence (XAI). The rapid growth in the XAI research outputs of the last decade is prominently due to the fast increase in the popularity of machine learning (ML), particularly of deep learning (DL) models, because of the astonishing levels of prediction accuracy that they can reach [1]. These models are nowadays applied in several types of knowledge and business areas, spanning from autonomous vehicles [2] to games [3] and including criminal justice, healthcare [1] and battlefield simulations [4], just to mention a few. Unfortunately, most of the ML and DL models are considered as a "black-box" by scholars, and more so by the lay public, because of their complex, non-linear underlying structures that make them opaque and unintelligible. This opacity has created the need for XAI architectures, mainly motivated by three reasons [4]: (i) the request to increase the transparency of the models; (ii) the necessity to allow humans to interact with them; and (iii) the demand for the trustworthiness of their inferences. This has led to the

development of a plethora of domain-dependent and context-specific methods for dealing with the interpretation of ML models and the formation of explanations for humans. This trend is far from being over, with an abundance of novel XAI methods that are scattered and need organisation. The goal of this article was to systematically review research works aimed at developing new methods for XAI, to make a comprehensive survey and define a novel classification system of a larger scope built on the efforts of other scholars to organise the vast number of XAI methods proposed so far [5]. The analysis of more than 200 scientific articles exposed a research gap in the literature. None of the existing classification systems considered the format of the explanations generated by the XAI methods. Thus, the system proposed in this manuscript includes this brand-new dimension, the format of explanations, in harmony with other categories. This new dimension enhances the practical character of the proposed system as it allows scholars and practitioners to select the most relevant XAI method from the most suitable type of explanation for the problem at hand. The conceptual framework at the basis of the proposed system is represented in Figure 1. Most of the XAI methods focus on interpreting and making the entire process of building an AI system transparent, from the inputs to the outputs via the application of a learning approach to generate a model. The outcome of these methods are explanations that can be of different formats, such as rules, numerical, textual or visual information or a combination of the former ones (see Figure 2). The different formats of explanations are the natural consequence of the widespread application of AI-powered technologies that are utilised by different users in various fields to solve distinct problems [6–8]. As pointed out in [9], system designers, developers and AI practitioners find useful explanations that accurately reflect the logic implemented within a model. In this case, rule-based explanations represent a structured, compact yet comprehensible way to represent a set of logical instructions. On the other hand, end-users belonging to the lay public prefer reconstructive explanations that build a 'story', exposing which input features contribute the most to the model's prediction. Visual and textual explanations can tell why, for example, the image of an animal was assigned to a certain class in an intuitive way, such as "this image represents a penguin because it is white and black and it has a beak".

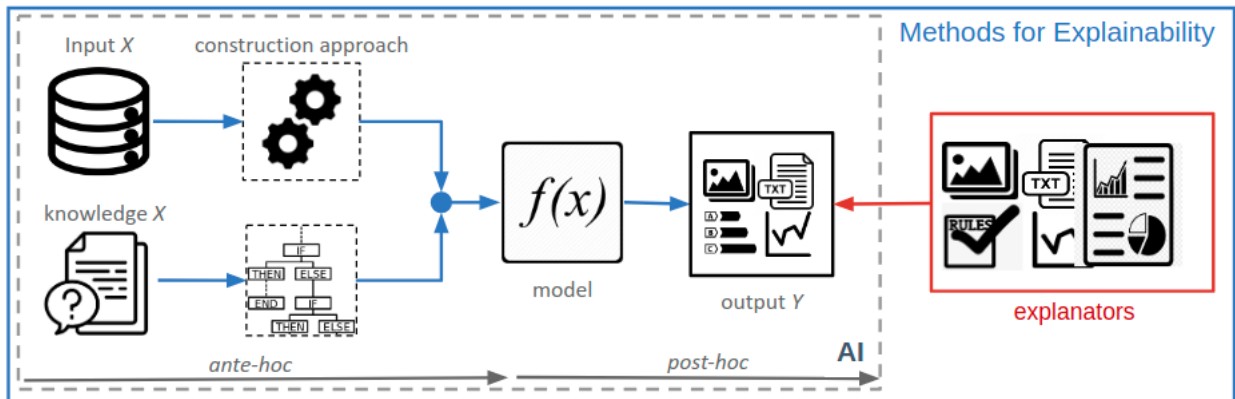

**Figure 1.** Diagrammatic view of how an explainable artificial intelligence (XAI) solution is typically constructed.

From several hundreds of research articles, more than 200 were considered for review by searching on Google Scholar papers related to "explainable artificial intelligence"; "explainable machine learning"; and "interpretable machine learning". Subsequently, the bibliographic section of these articles was thoroughly examined to retrieve further relevant scientific studies. The remainder of this paper is organised as follows. Section 2 provides a detailed description of the methodology followed to search for relevant research articles. Section 2.1 proposes a classification structure of the XAI methods describing top branches while Sections 3–7 expand this structure. Eventually, Section 8 concludes this review by summarising the gaps in the field highlighted by this survey, as well as suggesting future research work and challenges.

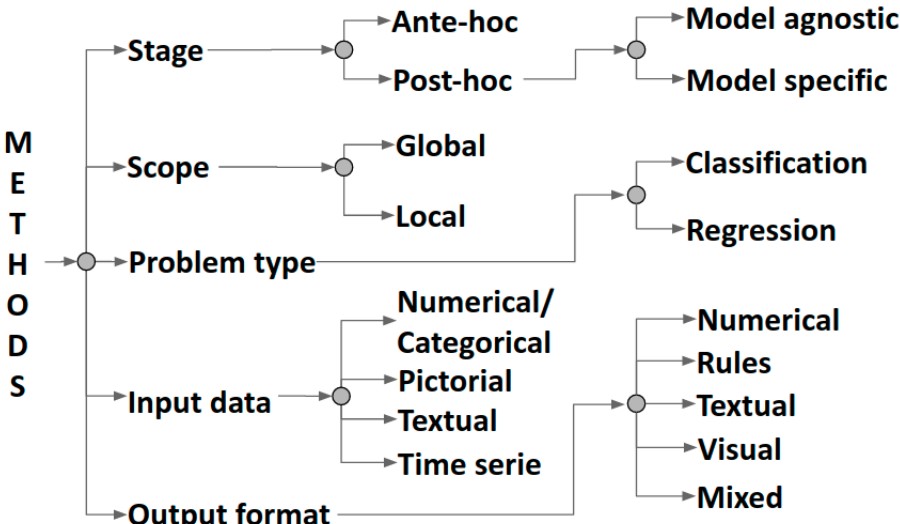

**Figure 2.** Classification of XAI methods into a hierarchical system.

## 2. Research Methods

The organisation of the scientific literature concerning new XAI methods within AI in a comprehensive and precise way and setting clear boundaries is far from being an easy task. This is the consequence of the multidisciplinary applications of these new techniques. Nonetheless, some constraints had to be defined, and the following publication types were excluded:

- Articles or technical reports that have not been peer reviewed;
- Scientific studies that applied existing XAI methods to specific problems, such as interpreting the forecasts made by DL models on images of cancers, and do not expand the XAI as a field. This exclusion was also necessary to drastically reduce the number of articles to something more manageable. Similarly, articles related to tutorials on XAI were discarded [10–13];
- Methods that could be employed for enhancing the explainability of AI techniques but that were not specifically designed for this purposes. For example, a considerable number of articles were devoted to methods designed for improving data visualisation or feature selection. These methods can indeed help researchers gain deeper insights into computational models, but they were not specifically designed for producing explanations.

Taking into account the above constraints, this review was carried out in two phases:

1. Google Scholar was queried to find articles discussing the explainability by using the following terms: *"explainable artificial intelligence"*; *"explainable machine learning"*; and *"interpretable machine learning"*. The search returned several thousands of results, but only the first ten pages contained relevant articles. Altogether, these searches provided a basis of around 170 peer-reviewed publications;
2. The bibliographic section of these articles was thoroughly reviewed. This led to the selection of further 50 articles whose bibliographic section was recursively analysed. This process was iterated until it converged and no more articles were found.

### 2.1. Classification of XAI Methods by Output Formats

More than 200 scientific articles were found that aimed to develop new XAI methods. Over time, researchers have tried to comprehend and unfurl the inner mechanics of data-driven and knowledge-driven models in various ways. From an examination of these articles, it was possible to identify the five main criteria for discriminating XAI methods. The first four have been already identified and analysed in the literature [5]. First, the *scope* of an explanation can be either *global* or *local*. In the former case, the goal is to make the entire inferential process of a model transparent and comprehensible as a whole.

In the latter case, the objective is to explain each inference of a model [14,15]. The second dimension refers to the *stage* at which a method generates explanations. *Ante hoc* methods aim to consider the explainability of a model from the beginning and during training to make it naturally understandable whilst still trying to achieve optimal accuracy [16–18]. *Post hoc* methods keep a trained model unchanged and mimic or explain its behaviour by using an external explainer at testing time [15,16,19,20]. The third dimension refers to the *problem type*. XAI methods can vary according to the underlying problem, either *classification* or *regression*. Finally, the mechanisms followed by a model to classify images can substantially differ from those used to classify textual documents, thus, the *input data* of a model (*numerical/categorical*, *pictorial*, *textual* or *times series*) can play an important role in constructing a method for explainability. Taking into account the articles examined in this systematic review, we propose an additional criteria, namely *output format* [21]. Similarly to input data, different circumstances can demand different formats of explanations to be considered by a method for explainability: *numerical*, *rules*, *textual*, *visual* or *mixed*. Figure 2 depicts in a graphical manner the structure of the proposed classification system of the XAI methods as a tree, whereas Figure 3 shows the distribution of the articles across its branches.

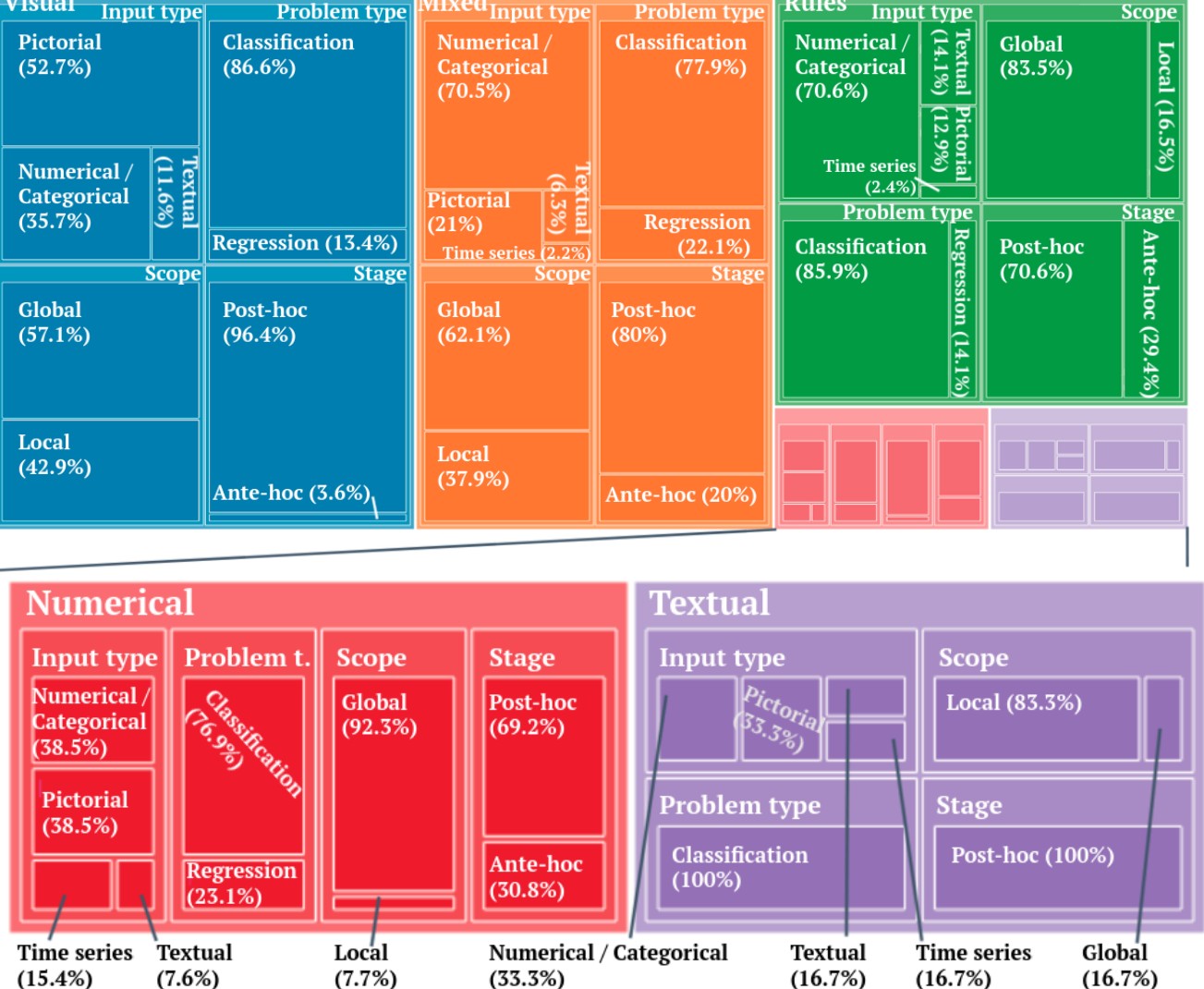

**Figure 3.** Distribution of XAI methods by output format across scope, stage, input type and problem type categories.

The output format category represents the main novelty of the proposed classification system that has not been considered yet in published surveys and it refers to the format

of the explanations produced by an XAI method. *Visual explanations* are probably the most natural way of communicating things and a very appealing manner to explain them. Scholars have analysed various types of visualisation tools to determine which ones are the most suitable for certain applications or meet the favour of scholars and practitioners. An example of these tools are heat-maps which highlight the specific areas of an image or specific words of a text that mostly influence the inferential process of a model by using different colours [22,23]. Visual explanations can also be used to illustrate the inner functioning of a model via graphical tools, such as the graphs proposed in [24], where each node is a layer of the network and the edges are the connections between layers. Another intuitive form of explanation for humans is *textual explanations*, natural language statements which can either be written or orally uttered. An example is a phrase "This is a Brewer Blackbird because this is a blackbird with a white eye and long pointy black beak" shown by an explainer of an image classification model [25]. *Rules* are a schematic, logical format, more structured than visual and textual explanations but still intuitive for humans. Rules can be in the form of "IF . . . THEN" statements with *AND/OR* operators and they are very useful for expressing combinations of input features and their activation values [26,27]. Technically, rules of these types employ symbolic logic, a formalised system of primitive symbols and their combinations (example: '$(Country = USA) \land (28 < Age \leq 37) \rightarrow (Salary > 50K)$' [28]). The parts before and after the $\rightarrow$ logical operator are, respectively, referred to as antecedent and consequent. Given this logic, rules can be implemented as fuzzy rules, linking one or more premises to a consequent that can be true to a degree, instead of being entirely true or false. This can be obtained by representing each antecedent and consequent as fuzzy sets [29]. Combining fuzzy rules with learning algorithms can become a powerful tool to perform reasoning and for instance, explain the inner logic of neural networks [30]. Similarly, the combination of antecedents and consequents can be seen as an argument in the discipline of argumentation, and a set of arguments can be put together in a dialogical structure by employing attacks, the link between arguments that model's conflicts [31,32]. Arguments and attacks form a complex structure but with high explanatory power, suitable for explaining the inner functioning of data-driven models. Explanations can also be constructed by only employing numerical formats as crisp values, vectors of numbers, matrices or tensors to highlight which input attributes and/or features of the model have the largest effect on the prediction of the output as in Probe [33] and concept activation vectors (CAVs) [34]. Numbers are capable of conveying information in a compact format, but they are perceived as dull and difficult to understand by many people. Eventually, some methods produce explanations based on a combination of the other four formats in the attempt to exploit their strengths and overcome their weaknesses, as performed by Image Caption Generation with the Attention Mechanism which jointly employs visual and textual explanations [35].

The following sections try to succinctly describe the XAI methods found during this systematic review grouped by the formats of the explanations generated by them. This is accompanied by tables for further classifying them according to their stage, scope, problem type and input data in alphabetic order. Given the large number of methods found for each output format, it was decided to further group them according to which learning approaches they can be applied to. Note that some of the XAI methods found in this study were designed to expose the logic of learning algorithms based on rules, such as decision trees, however the output explanations are not themselves rules but other formats, such as text or a graph. These methods can be found in the sections related to the appropriate explanation format. An example is a method called Feature Tweaking [36] which explains the logic of random forests, itself a learning algorithm built with rules, but it returns a numeric explanation corresponding to the size of a linear shift that must be applied to an input instance to be classified under another class. Thus, this method is classified under numeric explanations for ensemble algorithms. Indeed, the post hoc XAI methods can be further divided into *model-agnostic* and *model-specific* methods [5]. The former methods do not consider the internal components of a model, such as weights, therefore they can

be applied to any learning approach. The latter methods are instead limited to specific classes of models. For instance, the interpretation of the weights or activation values of a neural network model is specific to this learning approach (neural network) [14]. Model agnosticism and specificity do not apply to ante hoc methods because their goal is to make the functioning of a model transparent, so they are intrinsically model-specific [16]. Figure 4 shows the distribution of the scientific articles across the output formats and learning approaches.

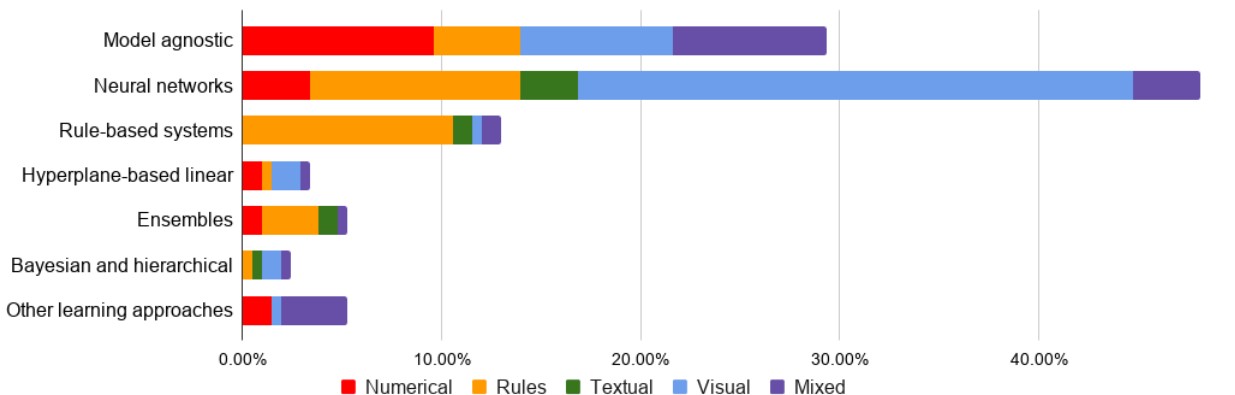

**Figure 4.** Distribution of the scientific articles of the XAI literature split by output format and learning approach.

## 3. Numeric Explanations

### 3.1. Model Agnostic XAI Methods

A few model agnostic XAI methods produce numerical explanations (see Table 1 and Figure 5 for some examples). Most of them focused on measuring the contribution of an input variable (or a group of them) with quantitative metrics. Distill-and-Compare [37] trains a transparent, simpler model, called a student, on the output obtained from a larger, complex model, considered as a teacher, to mimic its inferential process. In this study, the student model was constrained to be generalised additive models (GAMs) which easily allow estimating the contribution of each feature in a numerical format. Similarly, Shapley Additive Explanations (SHAP) [38] utilises additive feature attribution methods, which are basically linear combinations of the input features, to build a model which is an interpretable approximation of the original model. The authors recently proposed a modified version of SHAP, called TreeExplainer [39], specific for tree-based ML models such as those trained with random forests and gradient boosting. The SHAP algorithm is based on the assumption of features' independence, but [40] proposed a way to improve it by relaxing this assumption. Alternatively, Ref. [41] proposed to combine the Shapley values, which assess the marginal contribution of each input feature to the model's predictions, with the Lorenz Zonoids decomposition, which can be seen as a generalisation of the receiver operating characteristic (ROC) curve in a multidimensional setting, to determine the relevant features.

Some XAI methods are based on an "input perturbation" approach, and generally speaking, they work by modifying the reported values of the variables of an input instance to cause a change in the model's prediction. Explain and Ime [42,43] assess, respectively, the contribution of a particular input variable or a set of variables by quantifying their effects on the predictions of a given model when they are varied through their range of values. Global Sensitivity Analysis (GSA) method [44,45] follows the same process to rank input features according to their contribution to the predictions. Refs. [46–50] proposed a method to explain the prediction of a model at the instance level, also based on the contribution of each feature estimated by comparing the model output when all the features are known and when one or more of them are omitted. The contribution is positive for the features that lead to the prediction towards a class, negative for those that push the prediction against a class, and zero when they do not have influence. Four methods, namely

Quantitative Input Influence (QII) functions [51]; Gradient Feature Auditing (GFA) [52]; Influence functions [53]; and Monotone Influence Measures [54], utilise influence functions to assess the contribution of each feature to certain predictions. An influence function is a classic technique from statistics measuring the sensitivity of a model to changes in the distributions of the independent variables [53]. The perturbation of the input can be done in different ways such as applying a constant shift (influence functions), obscuring parts of the input (GFA), rotating, reflecting or randomly assign labels to the input (monotone influence measures). Feature Importance [55] and Feature Perturbation [56] are also based on algorithms that modify subsets of the input features to find groups of interacting attributes used by different classifiers and determine the extent to which a model exploits such interactions.

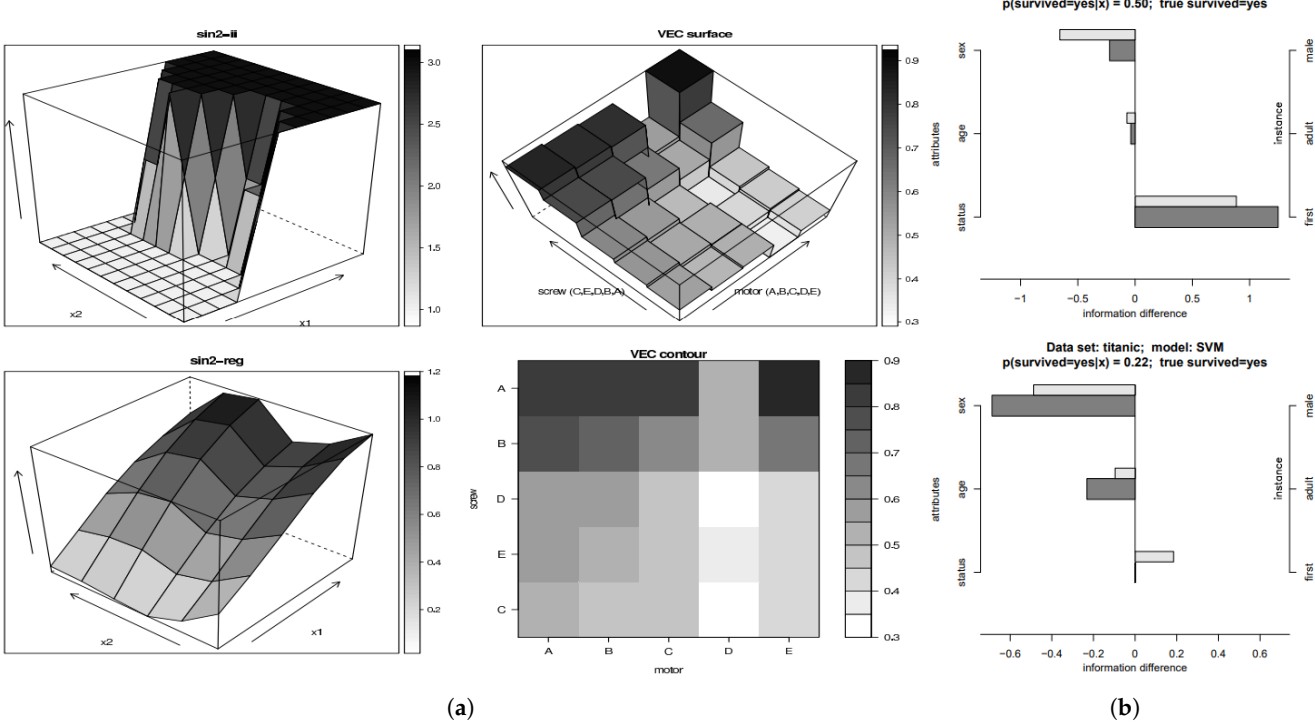

(**a**)                                                  (**b**)

**Figure 5.** Examples of numerical explanations generated by three model-agnostic XAI methods that highlight the contribution of the input features to the model's prediction and can be presented to the users as (**a**) surface charts (GSA [44]); or (**b**) bar plots (Explain and Ime [42]).

**Table 1.** Post hoc model-agnostic XAI methods generating numerical explanations, classified according to the type of problem (C: classification; R: regression), scope (G: global; L: local) and input data (NC: numerical/categorical; P: pictorials; T: textual; TS: time series) of the underlying model.

| Method for Explainability | Authors | Ref | Year | Scope | Problem | Input |
|---|---|---|---|---|---|---|
| Distill-and-Compare | Tan et al. | [37] | 2018 | G | C/R | NC |
| Explain and Ime | Robnik-Šikonja | [42,43] | 2008, 2018 | L | C | NC |
| Feature Contribution | Kononenko et al., Štrumbelj et al. | [46–48] | 2010, 2013, 2009 | L | C/R | NC |
| Feature Contribution | Štrumbelj et al. | [49,50] | 2008, 2010 | G | C/R | NC |
| Feature Importance | Henelius et al. | [55] | 2014 | G | C | NC |
| Feature Perturbation | Štrumbelj and Kononenko | [56] | 2014 | G | C/R | NC |
| GSA | Cortez and Embrechts | [44,45] | 2011, 2013 | G | C/R | NC |
| GFA | Adler et al. | [52] | 2018 | G | C/R | NC |
| Influence Functions | Koh and Liang | [53] | 2017 | G | C | P |
| Monotone Influence Measures | Sliwinski et al. | [54] | 2017 | L | C | P |
| QII functions | Datta et al. | [51] | 2016 | G | C | NC |
| SHAP | Lundberg and Lee, Janzing et al. | [38,40] | 2017, 2020 | G | C | P |
| Shapley–Lorenz–Zonoid Decomposition | Giudici and Raffinetti | [41] | 2020 | G | R | P |
| TreeExplainer | Lundberg et al. | [39] | 2020 | L | C | NC |

### 3.2. Model-Specific XAI Methods Based on Neural Networks

A few XAI methods produce pure numerical explanations for neural networks (see Table 2 and Figure 6). Concept activation vectors (CAVs) [34] separate the activation values of a hidden layer into two sets. The first set is relative to the instances belonging to a class of interest whereas the second set contains the activation values generated by the remaining part of the dataset. Then, it trains a binary linear classifier to distinguish the activation values of the two sets and compute directional derivatives on this classifier to measure the sensitivity of the model to changes in inputs towards the class of interest. This is a scalar quantity, calculated for each class over the whole dataset, which quantifies how important a user-defined concept is to classify the input instances in each class. For example, CAVs measure how sensitive the class "zebra" is to the presence of stripes in an input image. Probe [33] consists of a linear classifier fitted to a single feature learned by each layer of a deep neural network (DNN) to predict the original classes. The numerical explanations are the probability scores assigned by the probes to each class. Singular Vector Canonical Correlation Analysis (SVCCA) [57] returns the correlation matrix of the neurons' activation vectors, calculated over the entire dataset, of two instances of a given DNN trained separately. The first network's instance is obtained at the end of the training process, whereas the latter consists of multiple snapshots of the network during training. Causal importance [58] is computed by summing up the variations in the output when the values of a variable are perturbed, instance by instance, whilst all the other variables are kept fixed. The predictive importance of each variable is the absolute difference between the predictions made by a DNN with the original and the perturbed variables. Irrelevant variables are suppressed. The network is then trained with the relevant variables only and data are clustered according to their hidden layer representation by training an unsupervised Kohonen map on the matrix containing the activation values of each instance of a neuron/input pair. Finally, causal importance is measured on a cluster-by-cluster basis. Ref. [59] proposed to measure the contextual importance and contextual utility of input on the output variable. The former metric is the ratio between the range of the output values covered by varying a variable throughout its range of values and the whole output space. For example, a neural network was trained to predict the price of a car over a set of variables, among which there is the engine size. By varying the engine size alone, the price varies only within a certain range. Contextual utility represents the position of the actual output within contextual importance. The price of cars with big engines, produced by the same manufacturer, are towards the upper end of the manufacturer's price range. LAXCAT [60] identifies the features of input time-series data as well as the time intervals deemed relevant by a CNN to determine the output class. The CNN is coupled with a variable attention module that assigns weights to the input features according to their relevance and a temporal attention module that identifies the time intervals over which the features selected by the previous module inform the model's prediction. Finally, Recurrent Lexicon Network (RELEXNET) [61] combines the transparency of lexicon-based classifiers with the accuracy of recurrent neural networks (RNNs). Lexicons are linguistic tools for classification and feature extraction which consist of a list of terms weighted by their strength of association with a given class. RELEXNET uses lexicons as inputs of a naive gated RNN which returns the probability that the input belongs to a certain class.

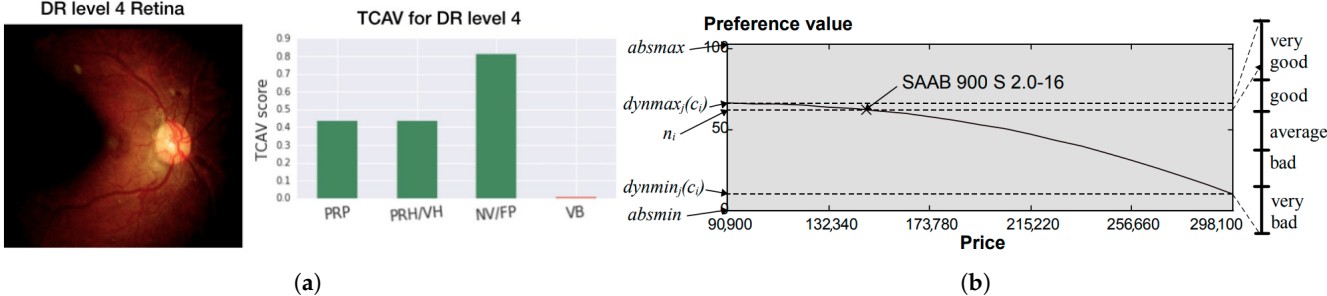

**Figure 6.** Examples of numerical explanation generated by a method for the explainability of neural networks showing the contribution of the most relevant features to the network's predictions: (**a**) Concept Activation Vectors [34]; and (**b**) contextual importance and utility [59].

**Table 2.** XAI methods for neural networks generating numerical explanations, classified according to the stage (AH: ante hoc; PH: post hoc), the type of problems (C: classification; R: regression), scope (G: global; L: local) and input data (NC: numerical/categorical; P: pictorials; T: textual; TS: time series) of the underlying model.

| Method for Explainability | Authors | Ref | Year | Stage | Scope | Problem | Input |
|---|---|---|---|---|---|---|---|
| Causal Importance | Féraud and Clérot | [58] | 2002 | PH | G | C | NC |
| CAVs | Kim et al. | [34] | 2018 | PH | G | C | P |
| Contextual Importance and Utility | Främling | [59] | 1996 | PH | G/L | C | NC |
| LAXCAT | Hsieh et al. | [60] | 2021 | PH | G | C | TS |
| Probes | Alain and Bengio | [33] | 2017 | PH | G | C | P |
| RELEXNET | Clos et al. | [61] | 2017 | AH | G | C | T |
| SVCCA | Raghu et al. | [57] | 2017 | PH | G | C/R | P |

### 3.3. Other Model-Specific XAI Methods

Scholars proposed other XAI methods generating numeric explanations for ML models that are not strictly based on neural networks (see Table 3).

#### 3.3.1. Ensembles

Feature Tweaking [36] modifies a variable (or a set of variables) of an input instance by applying a linear shift, capped to a global tolerance value until all the trees in the ensemble assign it to another target class. The delta between the original and the tweaked value represents the "tweaking cost" required to move the instance into the target class. Random forest model and sample explainer (RFEX) [62] returns numerical explanations, formatted as tables, of the predictions made by random forests in binary classification problems. The table contains the features of the dataset ranked according to their predictive power, measured by their mean decrease in accuracy, cumulative $F_1$ score and Cohen distance. The Cohen distance indicates the degree of separation in the feature values between the two output classes. It is the absolute difference of the average feature values' relative calculated over the samples belonging to the two classes divided by the feature's standard deviation.

#### 3.3.2. Support Vector Machines

Important support vectors and border classification [63] are two methods for providing insight into local classifications produced by an SVM. The former reports the most influential support vectors for the final classification of a particular data instance, thus determining the closest samples to the test point belonging to the same class. The latter determines which features of a testing instance would need to be altered (and by how much) to be classified on the separating surface between two classes. This provides a measure of the cost required to change a model's prediction. Weighted linear classifier [64] generates weighted linear SVM classifiers on random hyperplanes to obtain models whose accuracy is comparable to that of a non-linear SVM classifier and whose results can be readily visualised being projected on separating hyperplanes and decision surfaces. These projections are considered as a sort of explanation.

**Table 3.** XAI methods for data-driven approaches generating numerical explanations, classified according to the construction approach (learning algorithm), stage (AH: ante hoc; PH: post hoc), type of problems (C: classification; R: regression), scope (G: global; L: local) and input data (NC: numerical/categorical; P: pictorial; T: textual; TS: time series).

| Method for Explainability | Authors | Ref | Year | Construction Approach | Stage | Scope | Problem | Input |
|---|---|---|---|---|---|---|---|---|
| Feature Tweaking | Tolomei et al. | [36] | 2017 | Ensembles | PH | L | C | NC |
| Important Support Vectors and Border Classification | Barbella et al. | [63] | 2009 | SVM | PH | L | C | NC |
| RFEX | Petkovic et al. | [62] | 2021 | Ensembles | PH | G | C | NC |
| Weighted Linear Classifier | Caragea et al. | [64] | 2003 | SVM | PH | G | C | NC |

### 3.4. Self-Explainable and Interpretable Methods

Naturally interpretable models, sometimes referred to as "white-box models", are inherently "ante hoc" (see Table 4). Their output format depends on their architecture and input format. Gaussian process regression (GPR) [65] is a non-parametric regression algorithm which is interpretable because the weights assigned to each feature provide a measure of its relevance. Oblique Treed Sparse Additive Models (OT-SpAMs) [66] divide the input feature spaces into regions with sparse oblique tree splitting and assign local sparse additive predictive models to individual regions. Supersparse linear integer model (SLIM) [67] generates a scoring system from an input dataset by assigning a score to each variable. These scores are multiplied by a set of coefficients inferred from the training dataset and then added, subtracted, and/or multiplied to make a prediction. The scores are generated by solving a discrete optimisation problem that minimises the 0–1 loss to reach a high level of accuracy, regularises a $\ell_0 - penalty$ to encourage a high level of sparsity and constrains coefficients to a set of user-defined meaningful and intuitive values.

**Table 4.** Ante hoc XAI methods generating white-box models generating numerical explanations, classified according to the type of problem (C: classification; R: regression), scope (G: global; L: local) and input data (NC: numerical/categorical; P: pictorial; T: textual; TS: time series) of the underlying model.

| Method for Explainability | Authors | Ref | Year | Scope | Problem | Input |
|---|---|---|---|---|---|---|
| GPR | Caywood et al. | [65] | 2017 | G | R | TS |
| OT-SpAMs | Wang et al. | [66] | 2015 | G | C | NC |
| RELEXNET | Clos et al. | [61] | 2017 | G | C | T |
| SLIM | Ustun et al. | [67] | 2014 | G | C | NC |

## 4. Rule-Based Explanations

### 4.1. Model Agnostic XAI Methods

A few model-agnostic XAI methods produce rule-based explanations by exploiting several rule-extraction techniques (see Table 5 and Figure 7 for examples of this format). Generally, these rules approximate a black-box model but have higher interpretability. The method presented in [68] extracts logical formulas as decision trees by combining split predicates along paths from inputs to predictions into logical conjunctions and all the paths related to an output class into logical disjunctions. These rules can be analysed with logical reasoning techniques to extract information about the decision-making process. Similarly, Genetic Rule Extraction (G-REX) [69,70] employed genetic algorithms to generate IF-THEN rules with AND/OR operators. A genetic algorithm is also employed in GLocalX [71,72] to generate local rules that explain the prediction made by a classifier on a specific instance. The extracted rules exhibit the factual reasons of the classifier's predictions and suggest a set of counterfactuals consisting of changes to the instance features that lead to a different outcome. Afterwards, these local rules are hierarchically aggregated into a ruleset that covers the entire input space, thus representing a global explanation of the underlying model. Anchor [28] uses two algorithms to extract IF-THEN rules which highlight the features of an input instance, called "anchors", which are sufficient for a classifier to make

a prediction. In an analogical manner, the words "not bad" are often used in sentences expressing a positive sentiment. Thus, they can be considered anchors in sentiment analyses. These two algorithms, a bottom-up formation of and a beam-search for anchors, identify the candidate rules with the highest estimated precision over a dataset where precision is equal to the fraction of correct predictions. The first algorithm starts from an empty set of rules and adds, at each iteration, a rule for each feature predicate. On the other hand, the second one begins with a set containing all the possible candidate rules and then selects the best ones in terms of precision. Model Extraction [73] and Partition Aware Local Model (PALM) [74] utilise decision trees (DTs) to approximate complex models with the assumption that, as long as the approximation quality is good, the statistical properties of the complex model are reflected in the interpretable ones. End-users have the faculty to examine the DT's structure and determine whether the rules match intuition. Model Extraction generates DTs by using the Classification Furthermore, Regression Trees algorithm (CART) and trains them over a mixture of Gaussian distributions fitted to the input data using expectation maximisation. PALM uses a two-part surrogate model: a meta-model, constrained to be a DT, which partitions the training data, and a set of sub-models fitting the patterns within each partition. Mimic Rule Explanation (MRE) [75] returns a set of symbolic rules that behaves similarly to an underlying black-box model. First, the algorithm selects a set of prototype samples that covers the input space and records the label assigned by the model. Then, it perturbs each prototype to determine the maximum region where the class assigned by the model remains unchanged. The resulting ruleset consists of the Cartesian product of finite intervals limiting the regions surrounding the prototypes.

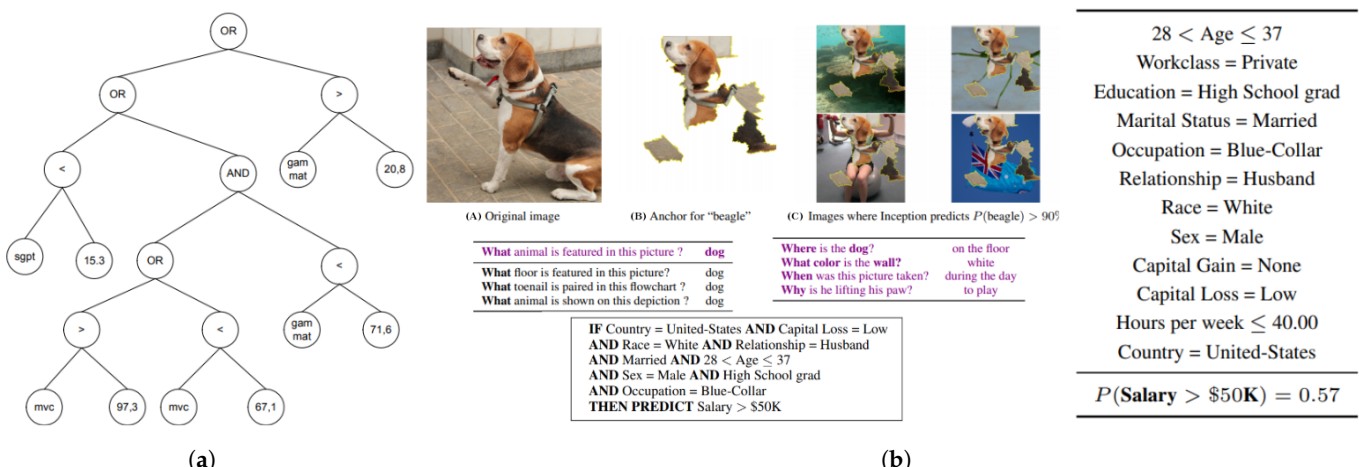

**(a)**          **(b)**

**Figure 7.** Examples of rule-based explanations generated by model-agnostic methods which can be visualised as: (**a**) G-REX [69], a decision tree, or (**b**) Anchor [28], a list of rules accompanied by textual and visual examples.

**Table 5.** Post hoc model-agnostic XAI methods generating rule-based explanations, classified according to the type of problem (C: classification; R: regression), the scope (G: global; L: local) and input data (NC: numerical/categorical; P: pictorials; T: textual; TS: time series) of the underlying model.

| Method for Explainability | Authors | Ref | Year | Scope | Problem | Input |
|---|---|---|---|---|---|---|
| Anchors | Ribeiro et al. | [28] | 2018 | G/L | C | T |
| Automated Reasoning | Bride et al. | [68] | 2018 | G | C | NC |
| GLocalX | Guidotti et al., Setzu et al. | [71,72] | 2019, 2021 | G/L | C | NC |
| G-REX | Johansson et al. | [69,70] | 2004 | G | C/R | NC |
| Model Extraction | Bastani et al. | [73] | 2017 | G | C/R | NC |
| MRE | Asano and Chun | [75] | 2021 | G | C | NC |
| PALM | Krishnan and Wu | [74] | 2017 | G | C/R | NC |

## 4.2. Model-Specific XAI Methods Based on Neural Networks

Several XAI methods are focused on rule-based explanations of the inferential process of neural networks, usually in the form of IF-THEN rules (see Table 6 and Figure 8 for some examples). Scholars divided these methods into three classes [76,77]: (I) *decompositional* methods, which work by extracting rules at the level of hidden and output neurons by analysing the values of their weights; (II) *pedagogical* methods, which treat an underlying neural network as a black-box and the extracted rules mimic the function computed by the network—where weights are not considered; and (III) *eclectic* methods, that are a combination of the decompositional and pedagogical ones. In contrast to previous sections, this section groups the XAI methods by employing the above three classes and not by the architectures of the neural networks.

*Decompositional methods*: Discretising Hidden Unit Activation Values by Clustering [78] generates IF-THEN rules by clustering the activation values of hidden neurons and replacing them with the cluster's average value. The rules are generated by examining the possible combinations in the outputs of the discretised network. Similarly, Neural Network Knowledge Extraction (NNKX) [79] produces binary decision trees from multi-layered feed-forward sigmoidal neural networks by grouping the activation values of the last layer and propagating them back to the input to generate clusters. Interval Propagation [30] is an improved version of Validity Interval Analysis (VIA) [80] to extract IF-THEN crisp and fuzzy rules. VIA finds a set of validity intervals such that the activation values of each unit (or a subset of units) of a DNN lie in these intervals. The precondition of each extracted rule consists of a set of validity intervals, and the output is a single target class. According to [30], sometimes VIA fails to decide whether a rule is compatible or not with the network. Additionally, the intervals are not always optimal. Interval Propagation overcomes these limitations by setting intervals to either the input or output and feed- or back-propagating them through the network. However, some neural networks require several crisp rules to approximate it and reach similar performance in terms of prediction accuracy. Then, Ref. [30] proposed to compact these crisp rules into fuzzy rules by using a fuzzy interactive operator that introduces the OR operators between rules. Discretised Interpretable Multi-Layer Perceptrons (DIMLPs) [27,77,81,82] returns symbolic rules from Interpretable Multi-Layer Perceptrons (IMLPs) which are convolutional neural networks (CNNs) where each neuron of the first hidden layer is connected to only an input neuron and has a step activation function while the remaining hidden layers are fully connected with a sigmoid activation function. In DIMLPs, the step activation function becomes a staircase function that approximates the sigmoid one. The rule extraction is performed after a max-pool layer by determining the location of relevant discriminative hyperplanes, which are the boundaries between the output classes. Their relevance corresponds to the number of points passing through each hyperplane as they move to a different class. An example of a ruleset generated with DIMLP from a neural network with thirty neurons, represented as $x_i$ with $i = 1, \ldots, 30$, in a unique hidden layer and three output neurons is: Rule 1: $(\neg x_3) \ (\neg x_8) \ (x_{17} > 0.0061) \ (x_{19} < 0.151) \ (x_{21} > 0.065) \ Class\_1$, Rule 2: $(x_{17} > 0.0061) \ (x_{21} < 0.065) \ Class\_2$, Default: $Class\_3$. Rule Extraction by Reverse Engineering (RxREN) [83] relies on a reverse engineering technique to trace back input neurons that cause the final result, whilst pruning the insignificant ones, and to determine the data ranges of each significant neuron in the respective classes. The algorithm is recursive and generates hierarchical rules where conditions for discrete attributes are disjoint from the continuous ones. Rule Extraction from Neural Network using Classified and Misclassified data (RxNCM) [84] is a modification of RxREN. It also incorporates the input instances correctly classified in the range determination process, not only the misclassified ones as RxREN does.

Most of the rule-based XAI methods are monotonic, which means they produce an increasing set of rules. However, sometimes adding new rules might lead to the invalidation of some conclusion inferred by other rules. A method that captures non-monotonic symbolic rules coded in the network was presented in [85]. The algorithm starts by partially ordering the vectors of a training dataset according to the activation values of the output layer. Then, it determines the minimum input point that activates an output neuron and creates a rule whose antecedents are based on the feature values of the selected instance. Thus, the expected set of rules has the following form: $L_1, \dots, L_n, \sim L_{n+1}, \dots, \sim L_m \to L_{m+1}$ where $L_i (1 \leq i \leq m)$ represents a neuron in the input layer, $L_{m+1}$ represents a neuron in the output layer, $\sim$ stands for default negation and $\to$ means causal implication. Finally, Refs. [86,87] proposed two algorithms that extract DTs from the weights of a DNN. The former method produces a soft DT trained by stochastic gradient descent using the predictions of a neural network and its learned filters to make hierarchical decisions on where to split the data and how to create the paths from the root to the leaves. The latter, which is designed only for image classification tasks, aims to explaining an underlying CNN semantically, meaning that the nodes of the tree should correspond to parts of the objects that can be named. To build such DTs, the network's filters are forced to represent object parts by a special modification of the loss function. The DT is then recursively built on the part/filter pairs on an image-by-image basis.

*Pedagogical methods*: Rule Extraction From Neural Network Ensemble (REFNE) [88] extracts symbolic rules, limited to only three antecedents, from instances generated by neural network ensembles. The algorithm randomly selects a categorical attribute and creates a rule if there is a value satisfying the condition that all the instances possessing such a value fall into the same class. If the condition is not satisfied, the algorithm selects another categorical attribute and examines all the combinations of the two attributes. When all the categorical attributes have been analysed, continuous attributes are considered and the process terminates when no more rules can be created. Continuous attributes are discretised and a fidelity evaluation mechanism checks that this process does not compromise the relationship between the attribute and the output classes. An alternative method to extract IF-THEN rules from trained neural network ensembles, called C4.5Rule-PANE [89], uses the C4.5 rule induction algorithm. To mimic the inferential process of the ensemble, the C4.5Rule-PANE extracts a ruleset from the modified version of the training dataset where the original labels are replaced by those predicted by the ensemble. The DecText method [90] extracts high fidelity DTs from a DNN. It sorts input instances by increasing order according to the values of each feature and split an input dataset by placing the cutpoint at the midpoint of the range. Then, DecText chooses the best partitions according to four criteria. The first, called SetZero, selects the most discriminative features of the target variable. The other three, SSE, ClassDiff and Fidelity, respectively, choose the feature which maximises the possibility that a single class dominating each partition is created, the quality of the partition and the fidelity between the DNN and the tree. TREPAN [91,92] induces a DT that, such as DecText, maintains a high level of fidelity to a DNN while being comprehensible and accurate. It queries an underlying network to determine the predicted class of each instance and selects the splits for each node of the tree by using the "gain ratio criterion" and by considering the previously selected splits that lie on the path from the root to that node as constraints. Tree Regularisation [93] consists of a penalty function of the parameters of a differentiable DNN which favours models whose decision boundaries can be approximated by small binary DTs. It finds a binary DT that accurately reproduces the network's prediction and measures its complexity as the "average decision path length". It then maps the parameter vector of each candidate network to an estimate of the average-path-length and chooses the shortest one. Word Importance Scores [94] visualises the importance of specific inputs for determining the output of an LSTM. By searching for phrases with consistently high importance scores, the method extracts simple phrase patterns of one to five words. To concretely validate these patterns, they are inputted to a rule-based classifier which approximates the performance

of the original LSTM. Iterative Rule Knowledge Distillation [95] and Symbolic Logic Integration [96] are the only ante hoc methods producing rule-based explanations for DNNs. The former combines DNNs with declarative first-logic rules to allow integrating human knowledge and intentions into the networks via an iterative distillation procedure that transfers the structured information of logic rules into the weights of DNNs. This is achieved by forcing the network to emulate the predictions of a rule-based teacher model and evolving both models throughout the training. The latter instead encodes symbolic knowledge in an unsupervised neural network by converting background knowledge, in the form of propositional IF-THEN rules and first-order logic formulas, into confidence rules which can be represented in a restricted Boltzmann machine.

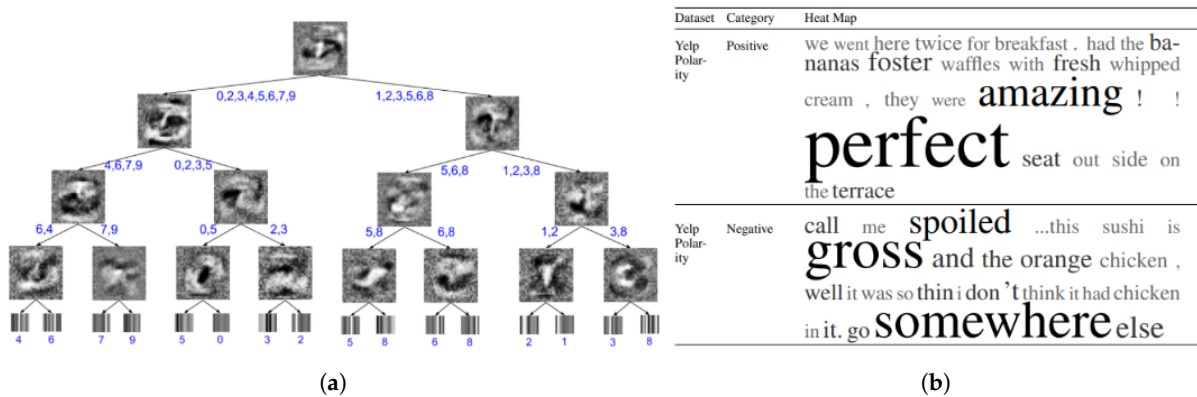

(**a**)                                                                (**b**)

**Figure 8.** Examples of rule-based explanations generated by XAI methods for neural networks and visualised as (**a**) decision trees (Decision Tree Extraction [86]), or (**b**) by showing the most relevant input (Word Importance Scores [94]).

**Table 6.** Post hoc XAI methods for neural networks generating rule-based explanations, classified according to the type of problem (C: classification; R: regression), scope (G: global; L: local) and input data (NC: numerical/categorical; P: pictorials; T: textual; TS: time series) of the underlying model.

| Method for Explainability | Authors | Ref | Year | Scope | Problem | Input |
|---|---|---|---|---|---|---|
| C4.5Rule-PANE | Zhou and Jiang | [89] | 2003 | L | C/R | NC |
| DecText | Boz | [90] | 2002 | G | C | NC |
| DIMLP | Bologna and Hayashi | [27,77,81,82] | 2017, 1998, 2018 | G/L | C | P; NC; T |
| Discretising Hidden Unit Activation Values by Clustering | Setiono and Liu | [78] | 1995 | G | C | NC |
| DT Extraction | Frosst and Hinton, Zhang et al. | [86,87] | 2017, 2019 | G | C | P |
| Interval Propagation | Palade et al. | [30] | 2001 | G | C | NC |
| Iterative Rule Knowledge Distillation | Hu et al. | [95] | 2016 | G | C | T |
| NNKX | Bondarenko et al. | [79] | 2017 | G | C | NC |
| REFNE | Zhou et al. | [88] | 2003 | G | C/R | NC |
| RxNCM | Biswas et al. | [84] | 2017 | G | C | NC |
| RxREN | Augasta and Kathirvalavakumar | [83] | 2012 | G | C/R | NC |
| Symbolic Logic Integration | Tran | [96] | 2017 | G | C/R | NC |
| Symbolic Rules | Garcez et al. | [85] | 2001 | G | C/R | NC |
| Tree Regularisation | Wu et al. | [93] | 2018 | G | C | NC |
| TREPAN | Craven and Shavlik | [91,92] | 1994, 1996 | G | C/R | NC |
| VIA | Thrun | [80] | 1995 | G | C/R | TS |
| Word Importance Scores | Murdoch and Szlam | [94] | 2017 | G | C | T |

### 4.3. Model-Specific XAI Methods Related to Rule-Based Systems

XAI was ignited by the interpretability problem of machine learning, particularly of the DL models. However, this problem existed even before the advent of neural networks. Many rules-based learning approaches that already existed were interpreted with ante hoc methods that act during the model training stage to make them naturally explainable (see Table 7 and Figure 9).

Ant Colony Optimisation (ACO) [97] follows a sequential covering strategy, one-rule-at-a-time or also known as separate-and-conquer, to generate unordered sets of IF-THEN classification rules which can be inspected individually and independently from the others, thus they are easier to be interpreted. At each step, ACO creates a new unordered set of rules and compares it with those of previous iterations. If the new set contains fewer rules or has a better prediction accuracy, it replaces the previous one. AntMinter+ [98] uses an iterative max–min ant system to construct a monotonic ruleset of IF-THEN rules, starting from an empty set, and allows the inclusion of domain knowledge via the definition of a directed acyclic graph representing the solution space. The nodes at the same depth in the graph represent the splitting values related to an input variable; the edges represent which values of the following variable can be reached from a node. A rule represents a path from the start to the end nodes. The algorithm stops when either a predefined percentage of training points is covered or when the addition of new rules does not improve the accuracy of the classifier. AntMinter+ can be combined with a non-linear SVM in a method called active learning-based approach (ALBA) to generate comprehensible and accurate rule-based models. Interpretable decision set [99] and the Bayesian rule lists (BRLs) [100–102] create unordered sets of IF-THEN rules. The former method is based on an objective function that simultaneously optimises accuracy and interpretability by learning short and non-overlapping rules that cover the whole feature space and pay attention to the small but important classes. BRLs produce a posterior multinomial distribution over the permutations of rules, starting from a large set of possible rules, to assess the probability of predicting a certain label from the selected rules. The prior is the Dirichlet distribution and the permutation that maximises that the posterior is included in the final decision set. Bayesian rule sets (BRSs) [103,104] are similar to BRL but it uses a Bernoulli distribution as a posterior, and a Beta distribution as a prior whose parameters can be adjusted by end-users by specifying the desired balance between the size and length of rules. First Order Combined Learner (FOCL) [105] inductively constructs a set of rules in terms of predicates where each clause body consists of a conjunction of predicates that cover some positive and no negative examples. The rules are displayed in a tree where the nodes are the predicates, the edges are the conjunctions and the leaves are the conclusions. Non-monotonic argumentation-based approaches for increasing explainability and dealing with conflictual information were proposed in [31,32,106]. They are based upon the concepts of defeasible arguments, in the form of rules, each composed of a set of premises, an inference rule and a conclusion as well as the notion of attacks between arguments to model's conflicts and the retraction of a final inference.

Four methods based on fuzzy reasoning to generate interpretable sets of rules that show the dependencies between inputs and outputs were presented in [107–110]. A multi-objective fuzzy Genetics-Based Machine Learning (GBML) algorithm [107] is implemented in the framework of evolutionary multiobjective optimisation (EMO) and consists of a hybrid version of the Michigan and Pittsburgh approaches. Each fuzzy rule is represented by its antecedent fuzzy sets as an integer string of fixed length and the resulting fuzzy rule-based classifier, consisting of a set of fuzzy rules, is represented as a concatenated integer string of variable length. Multi-Objective Evolutionary Algorithms-based Interpretable Fuzzy (MOEAIF) [110] instead consists of a fuzzy rule-based model engineered to classify gene expression data from microarray technologies. GBML and MOEAIF maximise the accuracy of rule sets, measured by the number of correctly classified training pattern, and minimise their complexity, measured by the number of fuzzy rules and/or the total number of antecedent conditions of fuzzy rules. The method in [108] is based on a five-step algorithm. First, it generates fuzzy rules that cover the extrema directly from the data. Second, it checks rule similarity to delete the redundant and inconsistent rules. Third, it optimises the rule structure using genetic algorithms based on a local performance index. Fourth, it performs further training of the rule parameters using the gradient-based learning method and deletes the inactive rules. Finally, it improves interpretability by using regularisation. The fourth method presented in [109] generates fuzzy rules by starting from

a set of relations and properties, selected by an expert, of an input dataset. It then extracts the most relevant ones by employing a frequent itemset mining algorithm. The authors did not provide a specific metric for evaluating the relevancy of a relation, but they suggested using "measures like the number of relations and properties in the antecedent or the value of their support".

Interpretable Classification Rule Mining (ICRM) [111] consists of a three-step evolutionary programming algorithm producing comprehensible IF-THEN classification rules, where comprehensibility is achieved by minimising the number of rules and conditions. First, it creates a pool of rules composed of a single attribute-value comparison. Second, it utilises evolutionary processes, designed to use only relevant attributes to discriminate a class from the others and improve the accuracy of the ruleset, based on the Iterative Rule Learning (IRL) genetic algorithm (also known as the Pittsburgh approach). IRL returns a rule per output class except for one class which is set as default. The third step optimises the accuracy of the classifier by maximising the product of sensitivity and specificity. Linear Programming Relaxation [112,113] extracts two-level Boolean rules in conjunctive normal form (AND-of-ORs) or disjunctive normal form (OR-of-ANDs). The first version uses a generalisation of a linear programming relaxation from one level to two-level rules whose objective function is a weighted combination of the total number of errors and features used in the rule. In a second version, the 0–1 classification error is replaced with the Hamming distance between the current rule and the closest rule that correctly classifies a sample instance. The main advantage of the explainability of the Hamming distance is that it avoids identical clauses in the ruleset, and thus repetitions, by training each clause with a different subset of input instances. Ref. [114] proposed to use constrained interval type-2 (CIT2) fuzzy sets to generate self-explainable rule-based classifiers. Transparent Generalised Additive Model Tree (TGAMT) [115] uses a CART-like greedy recursive search to grow a DT. Probabilistic sentential decision diagrams (PSDD) [116] can be described as circuit representations where each parameter represents a conditional probability of deciding the input variables and each node is either a logical AND gate with two inputs or a logical OR gate with an arbitrary number of inputs. The PSDD structure can be visualised as a binary tree.

if hemiplegia **and** age $> 60$ **then** *stroke risk* 58.9% (53.8%–63.8%)
**else if** cerebrovascular disorder **then** *stroke risk* 47.8% (44.8%–50.7%)
**else if** transient ischaemic attack **then** *stroke risk* 23.8% (19.5%–28.4%)
**else if** occlusion and stenosis of carotid artery without infarction **then** *stroke risk* 15.8% (12.2%–19.6%)
**else if** altered state of consciousness **and** age $> 60$ **then** *stroke risk* 16.0% (12.2%–20.2%)
**else if** age $\leq 70$ **then** *stroke risk* 4.6% (3.9%–5.4%)
**else** *stroke risk* 8.7% (7.9%–9.6%)

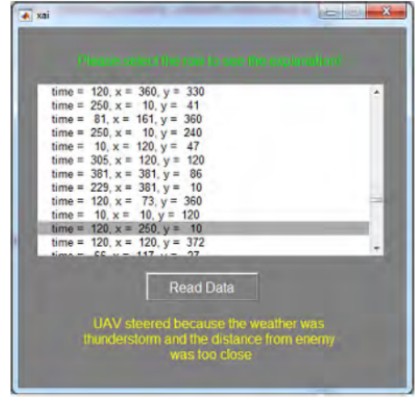

(**a**)                                                                                                (**b**)

**Figure 9.** Examples of rule-based explanations generated by ante hoc XAI methods aiming to make rule-based inference systems transparent by construction: (**a**) Bayesian rule lists [102]; and (**b**) fuzzy inference systems [117].

**Table 7.** Ante hoc XAI methods for rule-based inference systems generating rule-based explanations, classified according to the type of problem (C: classification; R: regression), scope (G: global; L: local) and input data (NC: numerical/categorical; P: pictorials; T: textual; TS: time series) of the underlying model.

| Method for Explainability | Authors | Ref | Year | Scope | Problem | Input |
|---|---|---|---|---|---|---|
| ACO | Otero and Freitas | [97] | 2016 | G | C | NC |
| AntMiner+ and ALBA | Verbeke et al. | [98] | 2011 | G | C | NC |
| Argumentation | Rizzo and Longo | [31,32] | 2018 | G | C | NC |
| Argumentation | Zeng et al. | [106] | 2018 | G | C/R | P |
| BRL | Letham et al. | [100–102] | 2012, 2013, 2015 | G | C | NC |
| BRS | Wang et al. | [103,104] | 2016, 2017 | G | C | NC |
| CIT2 fuzzy sets | D'Alterio et al. | [114] | 2020 | G | C | NC |
| Interpretable Decision Set | Lakkaraju et al. | [99] | 2016 | G | C | NC |
| FOCL | Pazzani | [105] | 1997 | G | C | NC |
| Fuzzy logic | Pierrard et al. | [109] | 2018 | L | C | NC |
| Fuzzy system | Jin | [108] | 2000 | G | C | NC |
| GBML | Ishibuchi and Nojima | [107] | 2007 | G | C | NC |
| ICRM | Cano et al. | [111] | 2013 | G | C | NC |
| Linear Programming Relaxation | Malioutov et al., Su et al. | [112,113] | 2017, 2016 | G | C | NC |
| MOEAIF | Wang and Palade | [110] | 2011 | G | C | NC |
| PSDD | Liang and Van den Broeck | [116] | 2017 | G | C/R | NC |
| TGAMT | Fahner | [115] | 2018 | G | C | NC |

## 4.4. Other Model-Specific XAI Methods

Some methods generate rule-based explanations for learning approaches other than neural network and rule-based systems (see Table 8).

### 4.4.1. Ensembles

Five global methods for extracting a single DT from ensemble models were presented in [118–122]. In detail, Ref. [118] proposed a three-step algorithm to efficiently merge a set of DTs, each trained independently on distributed data, into a single tree. First, each DT is converted into a ruleset where each rule replicates a path from the root to a leaf and defines a region in the output space. All the regions are disjoint and they cover the entire feature space. Second, the regions are combined by using a line sweep algorithm which sorts the limits of each region and merges adjacent regions. Finally, a DT is extracted from the regions with an algorithm that mimics C5.0. inTrees [119] iteratively extracts, prunes and selects rules from a tree ensemble. The algorithm starts from the set of all the rules in the ensemble and excludes those covering a small number of instances. At each iteration, it selects the rule with the minimum error and shorter condition, and then it removes the instances satisfying this rule from the dataset and updates the initial ruleset according to the instances left while discarding rules that at this stage cover just a few, if not any instances and recalculating the error of the surviving rules. Ref. [120] uses the solution obtained from combining several hypotheses (or models) of the ensemble as an oracle, and it selects the single hypothesis that is most similar to the oracle. The similarity is measured according to three formal metrics: "*θ-measure*" which determines the probability that both classifiers agree; "*κ-measure*" which assesses the probability that two classifiers agree by chance; and "*Q-measure*" which assigns values between 0 and 1 to classifiers that correctly predict the same input instances and values between $-1$ and 0 to classifiers that commit errors on different instances. Ref. [121] creates a set of rule conjunctions representing the original random forest which is then hierarchically organised as a decision tree, whereas the method presented in [122] uses a divide-and-conquer algorithm analogous to the C4.5 algorithm. Factorised asymptotic Bayesian (FAB) [123] consists of a Bayesian model selection algorithm that simplified and optimised a tree ensemble. FAB estimates the model's parameters and the optimal number of regions of the input space (ensemble methods often split the input space into a huge number of regions) to derive a simplified model with appropriate complexity and prediction accuracy.

**Table 8.** Post hoc XAI methods for data-driven approaches generating rule-based explanations, classified according to the construction approach (learning algorithm), type of problem (C: classification; R: regression), scope (G: global; L: local) and input data (NC: numerical/categorical; P: pictorials; T: textual; TS: time series).

| Method for Explainability | Authors | Ref | Year | Construction Approach | Scope | Problem | Input |
|---|---|---|---|---|---|---|---|
| DT extraction | Andrzejak et al. | [118] | 2013 | Distributed DTs | G | C/R | NC |
| DT extraction | Ferri et al., Sagi and Rokach, Van Assche and Blockeel | [120–122] | 2002, 2020, 2007 | Ensembles | G | C | NC |
| EBI | Yap et al. | [124] | 2008 | Bayesian networks | G | C | NC |
| ExtractRule | Fung et al. | [26] | 2005 | Hyperplane-Based Linear Classifiers | G | C | P; NC |
| FAB inference | Hara and Hayashi | [123] | 2018 | Ensembles | G | C | NC |
| inTrees | Deng | [119] | 2018 | Ensembles | G | C/R | NC |

### 4.4.2. Support Vector Machines

ExtractRule [26] convert hyperplane-based linear classifiers, such as SVMs, into a set of non-overlapping symbolic rules which display, in a compact format, the inferential process of the underlying classifier. For example, a rule extracted from a classifier trained to distinguish between malign and benign tumours is "$(Cell\ Size \leq 3) \wedge (Bare\ Nuclei \leq 1) \wedge (Normal\ Nucleoli \leq 7) \implies mass\ is\ benign$". Each rule can be seen as a hypercube in the multidimensional space generated by the input variables with edges parallel to the axis. To define these hypercubes, each iteration of this algorithm is formulated as one of two possible optimisation problems. The first formulation seeks to maximise the volume covered by each rule whereas the second maximises the number of samples covered.

### 4.4.3. Bayesian and Hierarchical Networks

Explaining Bayesian Network Inferences (EBI) [124] produces a DT showing how the variables of a Bayesian network interact to make predictions and compensate missing and erroneous values. In detail, EBI explains the value of a target node in terms of its causal relationships with the influential nodes in the target's Markov blanket which include the target's parents, children and the children's other parents by working backwards from the target node.

## 5. Textual Explanations

Some scholars proposed post hoc XAI methods generating textual explanations for neural networks (see Table 9) and other learning algorithms.

**Table 9.** Post hoc XAI methods for neural networks generating textual explanations, classified according to the type of problem (C: classification; R: regression), scope (G: global; L: local) and input data (NC: numerical/categorical; P: pictorials; T: textual; TS: time series) of the underlying model.

| Method for Explainability | Authors | Ref | Year | Scope | Problem | Input |
|---|---|---|---|---|---|---|
| InterpNET | Barratt | [125] | 2017 | L | C | P |
| Most-Weighted-Path, Most-Weighted-Combination and Maximum-Frequency-Difference | García-Magariño et al. | [126] | 2019 | L | C | TS |
| Neural-Symbolic Integration | Bennetot et al. | [127] | 2019 | L | C | P |
| Rationales | Lei et al. | [128] | 2016 | L | C | T |
| Relevance and Discriminative Loss | Hendricks et al. | [25,129] | 2018, 2016 | L | C | P |

### 5.1. Model-Specific XAI Methods Based on Neural Networks

InterpNET [125] utilises the activation values of a DNN to generate natural language explanations of classifications done by external models and produces statements such as "This is an image of a Red Winged Blackbird because..." (see Figure 10). Three similar methods were proposed by [126]: Most-Weighted-Path, Most-Weighted-Combination and Maximum-Frequency-Difference. Most-Weighted-Path starts from the output neuron and

selects the corresponding input passing, layer-by-layer, via the neuron connected with the highest weight. Then, it auto-generates a natural language explanation indicating the most relevant feature for predicting the output category. Most-Weighted-Combination selects the two most-weighted input features. Maximum-Frequency-Difference retrieves, for each instance of the training dataset, its most similar cases. Then, it calculates the difference between the percentages of samples sharing or not the same output. The explanation is based on the input with the highest difference and is a statement such "the smart kitchen estimates that you are sad because you are eating chocolate, which is 50% more frequent in this emotional state than people in other emotional states". The integration of symbolic rules and neural networks to generate a natural language explanation of the network's predictions was analysed in [127]. The rules are extracted from a first neural network as a knowledge graph and then used to influence the learning process of a second network by modifying its hyperparameters. The rules and the network's predictions are fed into an automated reasoning system combined with a natural language generator. Rationales [128] justify the predictions made by DNNs in NLP tasks, such as sentiment analysis, by extracting pieces of the input text as justifications or rationales. These rationales must contain the words leading to the same prediction of the entire input text. They are selected via the combination of a "rationale generator" function, tagging all the words to be or not to be included in the rationale, and an "encoder" function that maps a string of words to a target class. Relevance and discriminative loss [25,129] generates textual explanations for an image classifier such as "The bird in the photo is a White Pelican because...". It consists of a CNN that extracts visual features from the images, such as colours and object parts, and two LSTMs that produce a description of each image conditioned on visual features. The training process aims to reduce two loss functions called, respectively, "relevance" and "discriminative", which assure that the generated sentences are both image relevant and category specific.

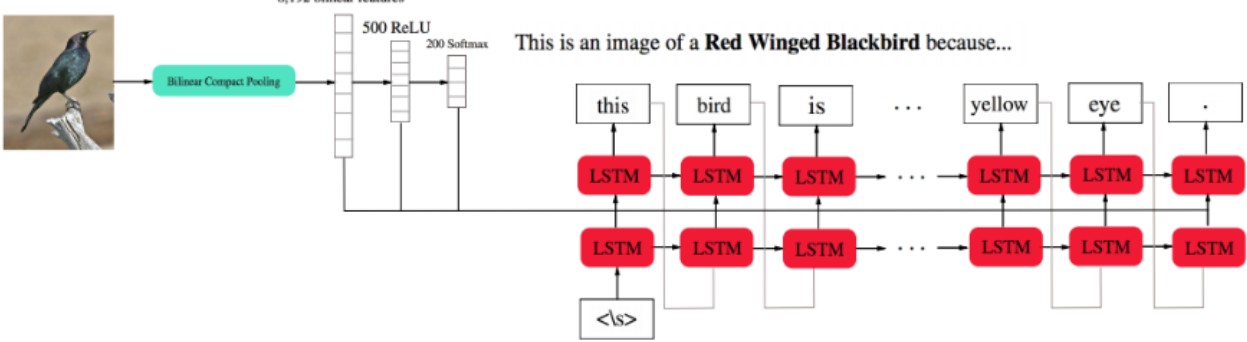

**Figure 10.** Examples of textual explanation generated by InterpNET [125], a XAI method for neural networks that utilises their activation values to extract the most significant input features and translates them into a statement.

### 5.2. Other Model-Specific XAI Methods

Table 10 summarizes the XAI methods designed for models based on rule-based, ensembles, bayesian and hierarchical networks.

### 5.2.1. Rule-Based Systems

Mycin [130], probably the first XAI method ever developed, is a backward chaining expert system. It consists of a knowledge basis of IF-THEN rules composed by an expert, a database of the facts that satisfy the condition part of the rules, and an inference engine that interprets these rules. It also includes a natural language interface that allows end-users to interact with the system by asking English questions. This system responds to them by using its inference engine and performing the reasoning involved in composing an answer. In detail, it searches for facts that match the condition part of the productions that match the action part of the question. This method allows the system to explain its reasoning and

final inferences by using AND/OR trees created during the production system reasoning process, thus showing an element of explainability. Similarly, the Sugeno-type fuzzy inference system [117] consists of an explicit declarative knowledge representation of the rules fired at the same time by a given input to produce a final inference. The system includes an explanatory tool that shows a numerical representation of the input variables, sets of co-fired rules and an English statement exposing the reasoning process. In an example taken from the application to an unmanned aerial vehicle (UAV) sent on a fight mission, an explanation is a statement such as "UAV aborted the mission because the weather was a thunderstorm and the distance from the enemy was too close".

**Table 10.** Post hoc XAI methods for data-driven approaches generating textual explanations, classified according to the construction approach (learning algorithm), type of problem (C: classification; R: regression), scope (G: global; L: local) and input data (NC: numerical/categorical; P: pictorials; T: textual; TS: time series).

| Method for Explainability | Authors | Ref | Year | Construction Approach | Scope | Problem | Input |
|---|---|---|---|---|---|---|---|
| DT Extraction | Alonso et al. | [131] | 2018 | Ensembles | L | C | NC |
| Discriminative Patterns | Gao et al. | [132] | 2017 | Ensembles | G | C | T |
| Fuzzy Inference Systems | Keneni et al. | [117] | 2019 | Rule-Based System | L | C | TS |
| Mycin | Shortliffe et al. | [130] | 1975 | Rule-based system | L | C | NC |
| Scenarios | Vlek et al. | [133] | 2016 | Bayesian networks | L | C | NC |

5.2.2. Ensembles

Ref. [131] combined an opaque learning algorithm (random forest), with a more transparent and inherently interpretable algorithm (decision tree). The opaque algorithm represents the oracle that searches for the most relevant output. A natural language generation approach composes a textual explanation for this output which is the interpretation of the inference process carried out by the correspondent decision tree if the outputs of both the learning algorithms coincide. Discriminative Patterns [132] interprets a random forest model that classifies sentences according to their contents. It extracts a ruleset that highlights discriminative sequential patterns of words or sentences that determine the predicted class.

5.2.3. Bayesian and Hierarchical Networks

An explanation method for interpreting Bayesian networks in terms of scenarios was proposed in [133]. Narrative approaches to reasoning with legal evidence, for instance, are based on the formulation of alternative scenarios which are subsequently compared according to two aspects: the relations with the evidence and the quality that depends on the completeness, internal consistency and plausibility of the scenario itself. The aim is to explain the content of a Bayesian network by reporting the modelled scenarios and evaluating their evidential support and quality.

## 6. Visual Explanations

### 6.1. Model Agnostic XAI Methods

Several model-agnostic XAI methods exploit graphical aids to explain the inner function of a model (see Table 11 and Figure 11 for examples of visual explanations). "Salient masks" are one of the most widely used graphical aids. Layer-Wise Relevance Propagation (LRP) [134] was developed as a solution to the problem of understanding image classification predictions by the pixel-wise decomposition of non-linear classifiers. In its general form, LRP assumes that the classifier can be decomposed into several layers of computation, and it traces back contributions of each pixel to the final output, layer by layer, to attribute relevance to individual inputs. The pixel contributions are visualised as heat-maps. Middle-Level Feature Relevance (MLFR) [135] is a variation of LRP. It returns the relevance values for a given set of middle-level features that consist of sets of pixels representing areas of the input images interpretable by humans. Spectral Relevance Analysis (SpRAy) [3] consists of spectral clustering on a set of LRP explanations to identify the typical and atypical decision

behaviours of an underlying data-driven model. For example, to explain a classifier trained on a dataset of images of animals, SpRAy produces an LRP heat-map for each image. Then, it checks if the heat-maps highlight the area representing the animal or if, for a specific animal, the classifier is focusing on other parts, such as the presence of a rider in case the animal is a horse. Image Perturbation [136] produces saliency maps by blurring different areas of the image and checking which ones most affect the prediction accuracy when perturbed. Similarly, Restricted Support Region Set (RSRS) Detection [137] visualises a set of size-restricted and non-overlapping regions of an image that are critical to its correct classification. The explanation consists of the original image with its critical regions greyed out. IVisClassifier [138] is based on linear discriminant analysis (LDA). It tries to reduce the dimension of the input data and produces heat-maps that give an overview of the relationship among the clusters in terms of pairwise distances between cluster centroids, both in the original and reduced dimensional spaces. The saliency detection method [139] uses a U-Net neural network trained to generate a saliency map, in a single forward pass, for any image and classifier received as inputs. The output map then highlights the parts of an image considered discriminative by the classifier.

Some methods use other visual aids, such as graphs and scatter-plots. Sensitivity Analysis [140] generates explanations that correspond to local gradients indicating which features of a sample must be modified to change its predicted label. The explanations are either scatter-plots or heat-maps of the gradient vectors showing the degree of sensitivity of the sample's features. Individual Conditional Expectation (ICE) plots [141] are line charts graphing the functional relationship between a predicted response and a feature for each observation when keeping all the other features fixed and varying the value of the feature under analysis. Two alternatives to ICE plots, called Partial Importance (PI) and Individual Conditional Importance (ICI) plots [142] show how changes in a feature affect model performance. ICI works at the local level by presenting changes for each observation. PI instead shows the point-wise average of all ICI curves across all observations, thus giving a global explanation. The importance of each feature is assessed with the Shapley Feature Importance measure that fairly distributes the model's performance among them according to their marginal contribution. Explanation Graph [143] is also based on the perturbations of the input features. It works by training a model on both the original and the perturbed data. Subsequently, it compares the original and perturbed input–output pairs to infer causal dependencies between the input and output. This method was tested across several word sequence generation tasks in natural language processing (NLP) applications. The perturbed input contains statements that are semantically similar to the originals but differ in some elements (words and punctuation) and their order. The inferred dependencies are displayed as graphs where the nodes contain the words of the original and perturbed inputs with their relative outputs and the edges represent the connections between them. A Worst-Case Perturbation [144] corresponds to the smallest perturbation of the input that leads to an incorrect answer with high confidence. This method was only applied to images, and the explanation consists of the perturbed images. Class Signatures [145] is an analytic interface that allows end-users to detect and interpret input–output relationships by presenting a mix of charts (line, bar charts, and scatter-plots) and tables organised in such a way that relationships become evident. Similarly, ExplainD [146] was designed to explain predictions made by classifiers that use additive evidence, such as linear SVMs and regressors. The explanatory graphs represent the contribution of each feature to the prediction and how the prediction changes when the value of a feature varies across their value ranges. Manifold [147] and MLCube Explorer [148] are two visual analytical tools that provide comparative analysis for multiple models. They also enable end-users to define the subsets of the input dataset using feature conditions to identify instances that generate erroneous results. The scope is to determine the reasons for these errors and to iteratively refine the performance of a model by using different graphical aids such as scatter-plots as well as bar and line charts.

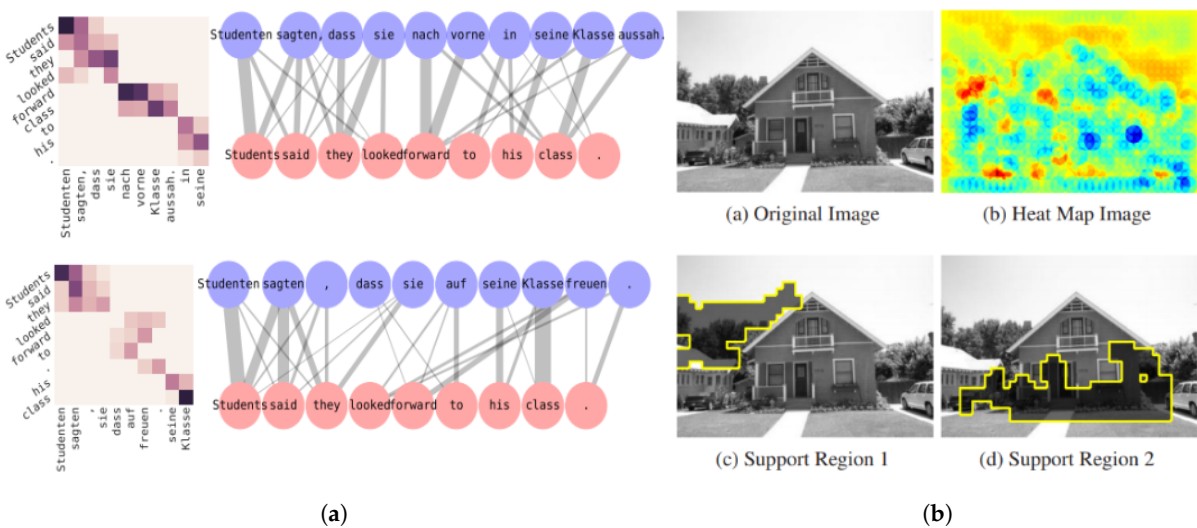

(**a**) (**b**)

**Figure 11.** Examples of visual explanations generated by model-agnostic methods such as (**a**) Explanation Graph [143] graphs; or (**b**) RSRS [137] restricted support regions and heat-maps.

**Table 11.** Post hoc model-agnostic XAI methods generating visual explanations, classified according to the type of problem (C: classification; R: regression), scope (G: global; L: local) and input data (NC: numerical/categorical; P: pictorials; T: textual; TS: time series) of the underlying model.

| Method for Explainability | Authors | Ref | Year | Scope | Problem | Input |
|---|---|---|---|---|---|---|
| Class Signatures | Krause et al. | [145] | 2016 | G | C/R | NC |
| ExplainD | Poulin et al. | [146] | 2006 | G | C | NC |
| Explanation Graph | Alvarez-Melis and Jaakkola | [143] | 2017 | L | C | T |
| Image Perturbation | Fong and Vedaldi | [136] | 2017 | L | C | P |
| ICE plots | Goldstein et al. | [141] | 2015 | G | C/R | NC |
| iVisClassifier | Choo et al. | [138] | 2010 | G | C | NC |
| LRP | Bach et al. | [134] | 2015 | L | C | P |
| Manifold | Zhang et al. | [147] | 2019 | G | C/R | NC |
| MLFR | Apicella et al. | [135] | 2021 | L | C | P |
| MLCube Explorer | Kahng et al. | [148] | 2016 | G | C | NC |
| PI and ICI plots | Casalicchio et al. | [142] | 2018 | G | C/R | NC |
| RSRS Detection | Liu and Wang | [137] | 2012 | L | C | T |
| Saliency Detection | Dabkowski and Gal | [139] | 2017 | L | C | P |
| Sensitivity Analysis | Baehrens et al. | [140] | 2010 | L | C | P; NC |
| SpRAy | Lapuschkin et al. | [3] | 2019 | G | C | P |
| Worst-Case Perturbations | Goodfellow et al. | [144] | 2015 | L | C | P |

*6.2. Model-Specific XAI Methods Based on Neural Networks*

A considerable portion of the reviewed scientific articles about new XAI methods is focused on interpreting deep neural networks (DNNs) with visual explanations. Given the large number of such methods proposed so far, it was decided to group them according to the type of visual explanation and the architecture of the underlying neural network.

6.2.1. Visual Explanations as Salient Masks

Most of the visual explanations of DNNs are in the form of salient masks (see Table 12 and Figure 12).

*Convolutional neural networks*: Class-Enhanced Attentive Response (CLEAR) [149] produces attention maps of the regions, along with their attentive levels, responsible for the correct classification of images by back-propagating the activation values of the output layer. GradCam [150] and DeepResolve [151] are two gradient ascent-based methods. GradCam uses the gradients of any target concept (say "dog" for instance) flowing into the

final convolutional layer to generate a heat-map highlighting the influential regions in the image for predicting that concept. GradCam returns the heat-maps produced by the last convolutional layer because the fully connected layers do not retain spatial information and it is expected that it has the best compromise between high-level semantics and detailed spatial information. DeepResolve computes and visualises intermediate layer feature maps that summarise how a network combines elemental layer-specific features to predict a specific class. Similarly, Integrated Gradients [152] attributes the prediction of a CNN or a RNN to specific parts of the input. The attribution is measured as the cumulative sum of the gradients of the classification function representing the network calculated at all points along the straight-line path from a baseline input (a black image or an empty text, for example) to a specific input instance. SmoothGrad [153] was designed to sharpen gradient-based sensitivity maps, which are often visually noisy as they highlight pixels that, to a human, seem randomly selected. It generates individual saliency maps of an image of interest by using other XAI methods such as GradCam, and then it considers the average of these maps. Alternatively, the joint mask method (JMM) [154] integrates two saliency masks to obtain faithful maps that focus on the essential parts of an object whilst removing noises. The first mask highlights the positive region of an input image that maximises the target class probability. The second mask attempts to find the negative region that significantly decreases the target class probability. The XAI method proposed in [155] attempts to highlight the salient features of the input images by averaging the activation values of a set of test images, thus identifying the relevant layer/filter pairs for every output class, and projecting them onto the image of interest as a saliency mask. Stacking with Auxiliary Features (SWAF) [156] utilises heat-maps generated by GradCam to interpret and improve stacked ensembles for visual question answering (VQA) tasks. VQA answers a natural language question about the content of an image by returning, usually, a word or phrase or, in this case, a heat-map highlighting the relevant regions. Guided BackProp and Occlusion [157] find what part of an input (pixels in images or words in questions) the VQA model focuses on while answering the question. Guided BackProp is another gradient-based technique to visualise the activation values of neurons in different layers of CNNs. It computes the gradients of the probability scores of predicted classes but restricts negative gradients from flowing back towards the input layer, resulting in sharper images showcasing the activation. Occlusion masks, or occludes, subsets of input (a region of the image or a word of the question), then forward propagate it through the VQA model and compute the change in the probability of the answer predicted with the original input. A similar method, Occlusion Sensitivity [158] maps those features considered relevant in the intermediate layers of a DNN by projecting the top nine activation values of each layer down to the input pixel space and masking the rest of the image. Net2Vec [159] maps semantic concepts to a corresponding individual CNN filter responses. It returns images that are entirely greyed out except in the region related to a semantic concept, such as the area representing the door of a building, for instance. The pixels of this region generate activation values that are above the 99.5th percentile of the distribution of all the activation values. Similarly, automated concept-based explanation (ACE) [160] extracts semantic concepts critical for determining the output class by segmenting the input images into objects and background as well as assigning to each segment an importance score based on the gradient of the prediction with respect to its pixels. Inverting representations [161] inverts the representations of images produced by the inner layers and projects them onto the input image as heat-maps. A representation can be thought of as a function of the image that characterises the image information. By reconstructing an approximate inverse function, it should be possible to reproduce the representations built by the layers. This method is based on the hypothesis that the layers only consider the relevant features and consist of a reconstruction problem solved by optimising an objective function with gradient descent. Similarly, Guided Feature Inversion [162] generates an inversion image representation consisting of the weighted sum between the original image and another noisy background image, such as a grey-scale image with each pixel set to an average

colour, a Gaussian white noise or a blurred image. The weights are computed to highlight the smallest area that contains the most relevant features and to blur out everything else.

Deep Learning Important Features (DeepLIFT) [163] calculates the importance scores of input features based on the difference between the activation of each neuron to a "reference activation" value, computed by propagating a "reference input" through the network. This represents a default or neutral input, such as a white image, chosen according to the problem at hand. Ref. [164] proposed to generate saliency maps by computing the first-order Taylor expansion of the function that links each pixel of an input image, thus representing the neural network, and assigns a probability score to each output class. Similarly, Ref. [165] analysed the use of Taylor decomposition for interpreting generic DNNs by decomposing the network's output classification into the contributions of its input elements and back-propagating them from the output to the input layer, which is then visualised as heat-maps. Receptive Fields [166] focus on visualising the input patterns, called precisely receptive fields, that are most strongly related to individual neurons by reconstructing these from the highest activation values of each layer. Feature Maps [167] and Prediction Difference Analysis [168] produce, respectively, feature- and heat-maps highlighting areas in an input image that gives evidence for or against a predicted class. Feature Maps utilises a loss function that pushes each filter in a convolutional layer to encode a distinct and unique object part, exclusive of the object class under analysis. Prediction Difference Analysis is instead based on Explain [42], which was designed to evaluate the contribution of a feature at a time. In this case, a feature should correspond to a pixel of the image, but the authors proposed to consider patches of pixels. The assumption is that the value of each pixel is highly dependent on the surrounding pixels. The patches are overlapping so that, ultimately, an individual pixel's relevance is calculated as the average relevance of the different patches it was in. PatternNet and PatternAttribution [169] measure the contribution of the input "signal" dimension, which is the part of the input that contains information about the output class, to the prediction as well as how good the network is at filtering out the "distractor", which is the rest of the input (like the image background). PatternNet yields a layer-wise back-projection of the estimated signal to the input space whereas PatternAttribution produces explanations consisting of neuron-wise contributions of the estimated signal to the classification scores. Relevant Features Selection [155] identifies the set of relevant layer/filter pairs by finding those that reduce the differences between the predicted and the actual labels at the minimum. This results in a relevance weight for every filter-wise response, internally computed by the network. Neural Information Flow (NIF) [170] utilises mutual information techniques for measuring information flows through the neural networks and for identifying the crucial pathways within and between its layers. A combination of a Neural Network and Case Base Reasoning (CBR) Twin-systems [171,172] maps the features' weights from the DNN to the CBR system to find similar cases from a training dataset that explain the prediction of the network of a new instance. To do so, the authors proposed the Contributions Oriented Local Explanations (COLE) technique, which assumes that the feature contributions to the model's predictions are the most sensible basis to inform CBR explanations. COLE uses saliency maps methods, such as LRP and DeepLift, to estimate these contributions. The OpenBox method [173] computes exact and consistent interpretations for the family of Piecewise Linear Neural Networks (PLNN) by transforming them into a mathematically equivalent set of linear classifiers.

*Recurrent neural networks*: Two studies proposed variations of LRP to extend its applications to DNNs with non-linearities, such as LSTMs with multiplicative interactions within their architecture [174] or networks with local renormalisation layers [175]. LRP with Relevance Conservation [174] consists of a strategy to back-propagate the relevance of the neurons in the output layer back to the input layer through the two-way multiplicative interactions between lower-layer neurons of the LSTM. The algorithm sets to zero the relevance related to the gate neuron and propagates the relevance of the source neuron only. Instead, LRP with Local Renormalisation Layers [175] is based on first-Taylor expansion for

non-linearities in the renormalisation layers. Compositionality [176] builds the meaning of a sentence from the meaning of single words and phrases. This method visualises compositionality in neural models trained for NLP tasks by plotting the salience value of each word as saliency maps. For instance, the word "hate" and "boring" in the phrase "I hate the movie because the plot is boring" can be considered the two most relevant ones in a sentiment analysis problem.

**Table 12.** XAI methods for neural networks generating visual explanations as salient masks, classified by stage (AH: ante hoc; PH: post hoc), type of problem (C: classification; R: regression), scope (G: global; L: local) and input data (NC: numerical/categorical; P: pictorials; T: textual; TS: time series).

| Method for Explainability | Authors | Ref | Year | Stage | Scope | Problem | Input |
|---|---|---|---|---|---|---|---|
| ACE | Ghorbani et al. | [160] | 2019 | PH | G | C | P |
| Average Activation Values | Mogrovejo et al. | [155] | 2019 | PH | L | C | P |
| CLEAR | Kumar et al. | [149] | 2017 | PH | L | C | NC |
| Compositionality | Li et al. | [176] | 2016 | PH | L | C | T |
| DeepLIFT | Shrikumar et al. | [163] | 2017 | PH | L | C | P; NC |
| Deep-Taylor Decomposition | Montavon et al. | [165] | 2017 | PH | G | C | P |
| DeepResolve | Liu and Gifford | [151] | 2017 | PH | G | C | NC |
| Feature Maps | Zhang et al. | [167] | 2018 | AH | L | C | P |
| GradCam | Selvaraju et al. | [150] | 2017 | PH | L | C | P |
| Guided BackProp and Occlusion | Goyal et al. | [157] | 2016 | PH | L | C | P |
| Guided Feature Inversion | Du et al. | [162] | 2018 | PH | L | C | P |
| Integrated Gradients | Sundararajan et al. | [152] | 2017 | PH | L | C | P |
| Inverting Representations | Mahendran and Vedaldi | [161] | 2015 | PH | L | C | P |
| JMM | Jung et al. | [154] | 2021 | PH | L | C | P |
| LRP w/Relevance Conservation | Arras et al. | [174] | 2017 | PH | L | C | T |
| LRP w/Local Renormalisation Layers | Binder et al. | [175] | 2016 | PH | L | C | P |
| Net2Vec | Fong and Vedaldi | [159] | 2018 | PH | G | C | P |
| NIF | Davis et al. | [170] | 2020 | PH | G | C | NC; P |
| Neural Network AND CBR Twin-Systems | Kenny and Keane, Kenny et al. | [171,172] | 2019, 2021 | PH | L | C | P |
| OcclusionSensitivity | Zeiler and Fergus | [158] | 2014 | PH | G | C | P |
| OpenBox | Chu et al. | [173] | 2018 | PH | G | C | P; NC |
| PatternNet, PatternAttribution | Kindermans et al. | [169] | 2018 | PH | L | C | P |
| Prediction Difference Analysis | Zintgraf et al. | [168] | 2017 | PH | L | C | P |
| Receptive Fields | He and Pugeault | [166] | 2017 | PH | G | C | P |
| Relevant Features Selection | Mogrovejo et al. | [155] | 2019 | PH | L | C | P |
| Saliency Maps | Simonyan et al. | [164] | 2014 | PH | L | C | P |
| SmoothGrad | Smilkov et al. | [153] | 2017 | PH | L | C | P |
| SWAF | Rajani and Mooney | [156] | 2017 | PH | L | C | P |

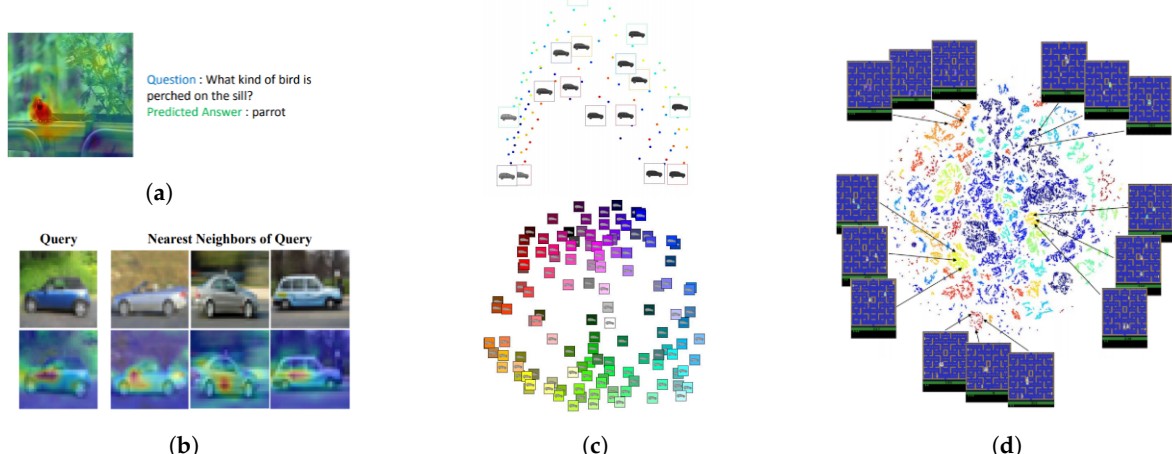

**Figure 12.** Examples of visual explanations generated by XAI methods for neural networks as salient masks (**a,b**) (Guided BP [157], Twin-systems [171]) and scatter-plots (**c,d**) (PCA [177], t-SNE maps [178]).

6.2.2. Visual Explanations as Scatter-Plots

Five methods explain DNNSs with scatter-plots (see Table 13 and Figure 12).

**Table 13.** Post hoc XAI methods for neural networks generating visual explanations such as scatter-plots, classified according to the type of problem (C: classification; R: regression), scope (G: global; L: local) and input data (NC: numerical/categorical; P: pictorials; T: textual; TS: time series).

| Method for Explainability | Authors | Ref | Year | Scope | Problem | Input |
|---|---|---|---|---|---|---|
| Cnn-Inte | Liu et al. | [179] | 2018 | G | C | P |
| Hidden Activity Visualisation | Rauber et al. | [180] | 2017 | G | C | P |
| Principal Component Analysis | Aubry and Russell | [177] | 2015 | G | C | P |
| t-SNE maps | Zahavy et al. | [178] | 2016 | G | C | NC |
| TreeView | Thiagarajan et al. | [181] | 2016 | G | C | P |

*Convolutional neural networks*: The Principal Component Analysis (PCA) method [177] measures the variation of CNN feature responses (or activation values) in the different layers to scene factors that occur in images such as object style, colour and lighting configuration. The Convolutional Neural Network Interpretation method (Cnn-Inte) [179] uses a two-level k-means clustering algorithm to group the activation values of the neurons in the hidden layers relative to each input feature. Then, a random forest algorithm is trained on each cluster. t-Distributed Stochastic Neighbour Embedding (t-SNE) maps [178] analyses Deep Q-networks (DQN) in reinforcement learning applications, particularly for agents that autonomously learn, for instance, how to play video games. This method extracts the neural activation values of the last DQN layer and applies t-SNE for dimensionality reduction and for generating cluster plots where each dot corresponding to a particular learning phase. Similarly, Hidden Activity Visualisation [180] uses t-SNE to visualise the projections of the activation values of the hidden neurons as a 2D scatter-plot with points coloured according to the class of the instances originating them. Finally, TreeView [181] consists of a scatter-plot representation of a DNN via the hierarchical partitioning of the feature space. Features are clustered according to the activation values of the hidden neurons. Each cluster comprises a set of neurons with a similar distribution of activation values across the whole training set.

6.2.3. Visual Explanations—Miscellaneous

Some methods use alternative visualisation tools (see Table 14 and Figure 13).

*Convolutional neural networks*: Generative Adversarial Network (GAN) Dissection [182] was designed to understand the inferential process of GANs at different levels of abstraction (from each neuron to each object) and the relationship between objects by identifying neurons (or groups of neurons) related to semantic classes (doors, for example). This method adds or removes these objects from the image and observes how the GAN network reacts to these changes. These reactions consist of a new version of the input image where other objects or background areas are modified by the GAN. For instance, if a door is intentionally removed from a building, the GAN might substitute it with a window or bricks. Score Deviation Map [183] is a perturbation-based method that inhibits specific areas of the input images according to the assigned semantic labels to determine which labels are relevant, irrelevant or distracting for the model. The explanations are maps of the input images assigning different colours to the various types of labels. Similarly, a recursive division method proposed by [184] hides rectangular parts of the input images of varying size to analyse their influence on the predictions of the underlying neural network.

Important Neurons and Patches [185] study the predictions of a DNN in terms of its internal features by inspecting information flow through the network. Given a trained network, important neurons are ranked according to two metrics, both measured over a set of perturbed images (each pixel is multiplied by a Gaussian noise): (I) the magnitude of the correlation between the neuron activation and the network output; and (II) the precision

of the activation of a neuron by selecting those neurons whose activation values were not significantly affected by the perturbations. The top N neurons are selected and their related image patches are determined by using a multi-layered deconvolutional network and enclosed in bounding boxes applied to input images. Two similar methods based on Activation Maximisation [186–188] modify the input images in such a way to maximise the activation of a given hidden neuron for each pixel. The modified images should provide a good representation of what a neuron is doing. Activation maps [189] shows what features activate the neurons in the penultimate layers. It assumes that the final prediction of a DNN is dominated by the most highly weighted neuron activations of this layer. Similarly, Ref. [190] proposed a three-step module that extracts the discriminative features of an image that most influence the model's predictions. First, a multi-attention CNN extracts a finite set of features. Then, an auto-encoder network is trained to generate prototypes for each part of the features, and finally, a predictor network that assigns a label per prototype. Gaussian Mixture Model (GMM) [191] computes the components of the activation values related to both training and test images to select the most activation-wise similar prototypes. Alternatively, Fractal View for Deep Learning [192] calculates the Euclidean distance of the layer-wise activation values of a set of input images and applies the Hilbert fractal curve to map similar images into a two-dimensional space. These curves are displayed in a pixel-grid where the colour of each pixel corresponds to the original output class of the input images.

A group of methods generates visual explanations in the form of graphs. Explanatory Graph [193] produces graphs from CNNs. Each node represents a "part pattern" corresponding to the peak activation in a layer related to a part of the input. Each edge connects two nodes in adjacent layers to encode co-activation relationships and spatial relationships between patterns. Similarly, Ref. [24] proposed to use data-flow graphs to visualise the structure of CNNs (but it applies to other DNN architectures, according to the authors) created and trained in Tensorflow. Symbolic Graph Reasoning (SGR) [194] consists of a layer added to CNNs which performs reasoning over a group of symbolic nodes whose outputs explicitly represent different properties of each semantic in a prior knowledge graph. To cooperate with local convolutions, each SGR is constituted by three modules: (a) a primal local-to-semantic voting module where the features of all symbolic nodes are generated by voting from local representations; (b) a graph reasoning module that propagates information over the knowledge graph to achieve global semantic coherency; (c) a dual semantic-to-local mapping module that learns new associations of the evolved symbolic nodes with local representations, and accordingly, enhances local features. Lastly, And–Or Graph (AOG) [195] grows a semantic AOG on a pretrained CNN. An AOG is a graphical representation of the problem's reduction to conjunctions (AND) and disjunctions (OR) of sub-problems (or sub-problems). It parses the part of the input images which corresponds to a semantic concept.

Some of the visual explanatory tools described thus far are employed to create interactive interfaces for the lay audience. For instance, Ref. [196] used saliency maps as the building blocks of such interfaces to explain the inferential logic of CNNs. ActiVis [197] unifies instance- and subset-level inspection with flowcharts showing how DNNs' neurons are activated by user-specified instances or instance subsets. Deep Visualisation Toolbox [198] depicts the activation values of every layer, produced while processing an image or video, as heat-maps and modifies the inputs via regularised optimisation methods to enable a better visualisation of the learned features by individual neurons. Deep View (DV) [199] measures the evolution of a DNN by using two metrics that evaluate the class-wise discriminability of the neurons in the final layer and the output feature maps. iNNvestigate [200] compares different XAI methods, namely PatternNet, PatternAttribution and LRP. Finally, $N^2VIS$ [201] produces interactive graphs representing the topology of feed-forward neural networks trained with evolutionary computation that allow end-users to adjust training parameters during adaptation and to see the results of this interaction.

*Recurrent neural networks*: Shifting from pictorial to textual inputs, Cell Activation Values [202] is a method of explainability for LSTMs. It uses character-level language

models as an interpretable test-bed for understanding the long-range dependencies learned by LSTMs by highlighting the sequences of relevant characters. LSTMVis [23] is an analysis tool for LSTMs that facilitates the understanding of their hidden state dynamics. It is based on a set of interactive graphs and heat-maps of relevant words. A user can select a range of text in the heat-maps, which results in the selection of a subset of hidden states visualised in a parallel coordinate plot where each state is a datum item, and time-steps are the coordinates. The tool then matches this selection to similar patterns in the dataset for further statistical analysis. Seq2seq-Vis [203] is similar to LSTMVis, but it focuses on sequence-to-sequence models, also known as encoder–decoder models, for the automatic translation of texts. Seq2seq-Vis allows interactions with trained models through each stage of the translation process intending to identify the learned pattern, detect errors and probe the model with counterfactual scenarios.

**Table 14.** XAI methods for neural networks generating miscellaneous visual explanations, classified according to the stage (AH: ante hoc; PH: post hoc), type of problem (C: classification; R: regression), scope (G: global; L: local) and input data (NC: numerical/categorical; P: pictorials; T: textual; TS: time series).

| Method for Explainability | Authors | Ref | Year | Stage | Scope | Problem | Input |
|---|---|---|---|---|---|---|---|
| Activation Maps | Hamidi-Haines et al. | [189] | 2019 | PH | L | C | P |
| Activation Maximisation | Erhan et al., Nguyen et al. | [186–188] | 2010, 2016 | PH | L | C | P |
| ActiVis | Kahng et al. | [197] | 2018 | PH | G | C/R | NC |
| AOG | Zhang et al. | [195] | 2017 | PH | G | C | P |
| Cell Activation Values | Karpathy et al. | [202] | 2016 | PH | G/L | C | T |
| Data-flow Graphs | Wongsuphasawat et al. | [24] | 2018 | PH | G | C/R | P; NC; T |
| Deep View | Zhong et al. | [199] | 2017 | PH | G | C/R | P |
| Deep Visualisation Toolbox | Yosinski et al. | [198] | 2015 | PH | G | C | P |
| Explanatory Graph | Zhang et al. | [193] | 2018 | PH | G | C | P |
| Fractal View for Deep Learning | Halnaut et al. | [192] | 2021 | PH | G | C | P |
| GMM | Stano et al. | [191] | 2020 | PH | L | C | P |
| GAN Dissection | Bau et al. | [182] | 2019 | PH | L | C | P |
| Important Neurons and Patches | Lengerich et al. | [185] | 2017 | PH | G | C | P |
| iNNvestigate | Alber et al. | [200] | 2019 | PH | L | C | P |
| LSTMVis | Strobelt et al. | [23] | 2018 | PH | G/L | C | T |
| N$^2$VIS | Streeter et al. | [201] | 2001 | PH | G | C/R | NC |
| Part Prototypes | Zhu et al. | [190] | 2021 | PH | G | C | P |
| Recursive Division Method | Gorokhovatskyi and Peredrii | [184] | 2021 | PH | L | C | P |
| Saliency Maps | Olah et al. | [196] | 2018 | PH | G/L | C | P |
| Score Deviation Map | López-Cifuentes et al. | [183] | 2021 | PH | L | C | P |
| Seq2seq-Vis | Strobelt et al. | [203] | 2018 | PH | L | C | T |
| SGR | Liang et al. | [194] | 2018 | AH | G | C/R | P |

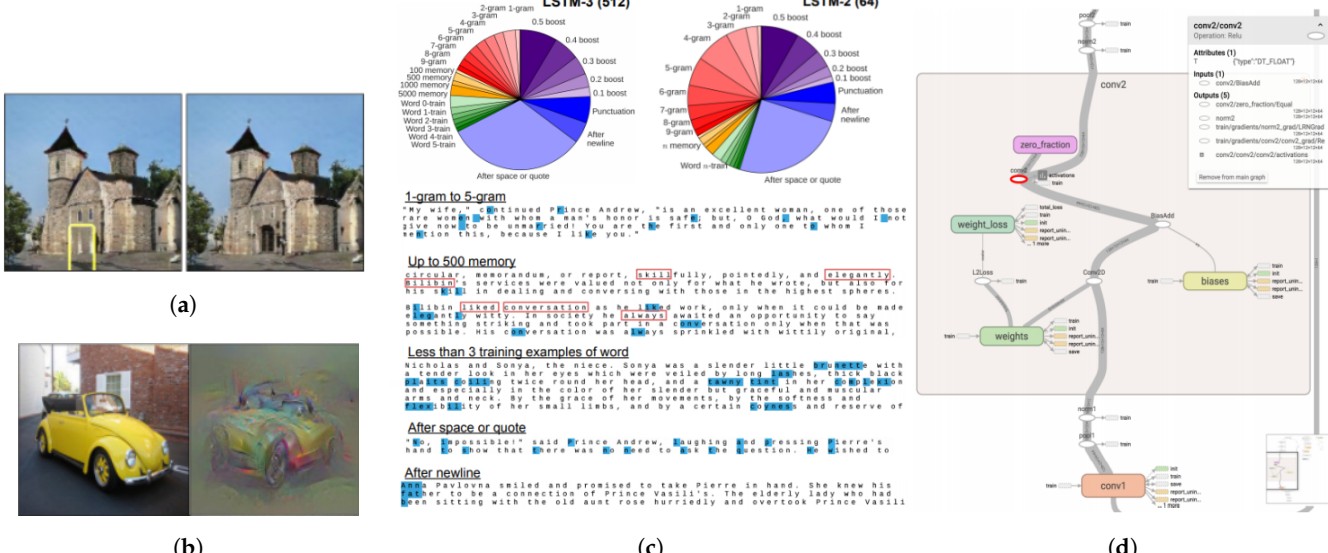

**Figure 13.** Examples of miscellaneous visual explanations generated by XAI methods for neural networks. Some methods modify the input images by removing parts to check the network's reaction—(**a**) GAN Dissection [182] or by maximising the activation of a given hidden neuron with respect to each pixel—(**b**) Activation Max [187]. Another alternative is to highlight the most relevant words of the input text—(**c**) Cell Activation [202] or to display the network's structure as a graph—(**d**) Data-Flow graphs [24].

### 6.3. Other Model-Specific XAI Methods

A few XAI methods generate visual explanations to interpret the logic of different learning algorithms (Table 15).

#### 6.3.1. Rule-Based Systems

Fuzzy Inference-Grams (Fingrams) [204] produces inference maps of sets of fuzzy rules. These maps depict the interaction between co-fired rules, support the detection of redundant or inconsistent rules and identify the most significant ones. This is achieved by using network scaling methods that simplify the maps while maintaining their most important relations. Fingrams also quantifies the comprehensibility of the ruleset, measured as the proportion of the co-fired rules. The assumption is that the larger the number of rules co-fired by a given input, the smaller the comprehensibility of the ruleset.

#### 6.3.2. Support Vector Machines and Naïve Bayesian-Driven Models

Self-Organising Maps (SOM) [205], an XAI method for SVMs, is an unsupervised network trained to detect, in a high-dimensional space of data, clusters of similar samples. It achieves this by projecting the input samples onto a 2-dimensional map while trying to preserve their topologies. It shows both data and the SVM models, providing an overview of the support vector decision surface. Refs. [206–208] introduced a method for automatically generating nomograms as visual explanations of the inferential mechanisms of SVM and naïve, Bayesian-driven models. A nomogram is a two-dimensional diagram designed to allow approximating the graphical computation of mathematical functions by showing a set of scales, one for each variable (dependent and independent) in an equation. By drawing a line connecting specific values of all the scales related to the independent variables, it is possible to calculate the value of the dependent variable from the intersection point between the line and the variable's scale. The advantages of the explainability of nomograms are the simplicity of presentation and clear display of the effects of individual attribute values.

### 6.3.3. Bayesian and Hierarchical Networks

Contribution propagation [209] is a per-instance method for hierarchical networks that explain which components of the input were responsible (and to what degree) for its classification. The central idea is that a node is relevant for the prediction if it was relevant to its parents, and its parents were also discriminative. The contribution of each input component is visualised as heat-maps.

**Table 15.** Post hoc XAI methods for data-driven approaches generating visual explanations, classified according to the construction approach (learning algorithm), type of problem (C: classification; R: regression), scope (G: global; L: local) and input data (NC: numerical/categorical; P: pictorials; T: textual; TS: time series).

| Method for Explainability | Authors | Ref | Year | Construction Approach | Scope | Problem | Input |
|---|---|---|---|---|---|---|---|
| Contribution Propagation | Landecker et al. | [209] | 2013 | Hierarchical networks | L | C | P |
| Fingrams | Pancho et al. | [204] | 2013 | Rule-based system | G | C | NC |
| Nomograms | Jakulin et al. | [206] | 2005 | SVM | G | C | NC |
| Nomograms | Možina et al. | [208] | 2004 | Naïve Bayes | G | C | NC |
| Self-Organising Maps | Hamel | [205] | 2006 | SVM | G | C | NC |
| VRIFA | Cho et al. | [207] | 2008 | SVM | G | C | NC |

### 6.4. Self-Explainable and Interpretable Methods

Two ante hoc methods produce visual outputs (Table 16). The first one, Symbolic Graph Reasoning (SGR) [194], was designed for CNNs and it is described in the previous subsection. Unsupervised Interpretable Word Sense Disambiguation [210] produces interpretable word sense disambiguation models that create clusters, or inventories, of words. For example, an inventory can collect all the words related to "furniture" (such as table, chair and bed). These words are clustered according to their co-occurrence and relative position in a text, where close words are assumed to be highly correlated. Their syntactic dependency is extracted from the Stanford Dependencies that represent grammatical relations between words in a sentence.

**Table 16.** Ante hoc XAI methods generating white-box models generating visual explanations, classified according to the type of problem (C: classification; R: regression), scope (G: global; L: local) and input data (NC: numerical/categorical; P: pictorials; T: textual; TS: time series) of the underlying model.

| Method for Explainability | Authors | Ref | Year | Scope | Problem | Input |
|---|---|---|---|---|---|---|
| Feature Maps | Zhang et al. | [167] | 2018 | L | C | P |
| SGR | Liang et al. | [194] | 2018 | G | C/R | P |
| Unsupervised Interpretable Word Sense Disambiguation | Panchenko et al. | [210] | 2017 | G | C | T |

## 7. Mixed Explanations

### 7.1. Model Agnostic XAI Methods

Many XAI methods produce numerical explanations along with graphical representations to make them more interpretable for lay people (see Table 17 and Figure 14 for examples of mixed explanations). Functional ANOVA decomposition [211] quantifies the influence of non-additive interactions within any set of input variables and depicts them with Variable Interaction Network (VIN) graphs where the nodes represent the variables, and the edges are the interactions. Combinatorial Methods [212], based on approaches derived from fault location in combinatorial testing, detect combinations of features that are present in input instances belonging to a certain class, but that are absent or rare in the rest of the input dataset. Justification Narratives [213] maps the essential values underlying a classification (identified with any feature selection method) to a semantic space that automatically produces narratives and shows them visually (as bar-charts reporting the estimated relevance value of each variable) or textually. Explainer [214] and Rivelo [215] are two user interfaces showing mixes of numerical, visual and textual explanations. Using TensorBoard (a visualisation tool developed by Google for machine learning), Explainer

produces an interactive graph view of a model where the nodes represent its components (such as inputs, parameters and outputs) accompanied by textual definitions, and the edges represent the relationships between them. Rivelo works exclusively with binary classification problems and binary inputs. It enables end-users to interactively explore a set of visual and textual instance-level explanations, such as lists of the most relevant input features (in words or images) and of instances that are correctly/wrongly classified.

Other mixed explanations consist of a selection of prototypes, input samples correctly predicted by the model that can be considered positive and iconic examples, or adversarial examples, samples misrepresented by the model suitable to generate contrastive explanations (concerning counterfactual and/or counter-intuitive events). This subset helps end-users understand the model by leveraging the human ability to induce principles from a few examples. These explanations are classified as mixed because their format depends on the nature of the input data. Bayesian Teaching [216] selects a small subset of prototypes that would lead the model to the correct inference as if trained on the overall dataset. Sequential Bayesian Quadrature (SBQ), in conjunction with Fisher kernels, selects salient training data points [217]. All the instances in a training dataset are embedded in the space induced by the Fisher kernels to quantify the closeness of pairs of instances which, if close enough, should be treated similarly by a model. The embedded instances are fed into SBQ, an importance-sampling-based algorithm estimates the expected value of a function under a distribution using discrete samples drawn from it. Set Cover Optimisation (SCO) [218] selects prototypes that capture the whole structure of the training data points in each class of the dataset. The prototypes cannot have another prototype of a different class in their neighbourhood and must be as few as possible. This leads to a set cover optimisation problem to be solved approximately with standard approaches such as, for instance, "linear program relaxation with randomized rounding". Neighbourhood-Based Explanations [219] is based on a Case-Based Reasoning (CBR) approach. It presents to end-users the entries of a training dataset that are the most similar to a novel input instance by employing Euclidean distance. Adversarial examples are instead used by C-CHVAE [220], Diverse Counterfactual Explanations (DiCE) [221], Evasion-Prone Samples Selection [222], Maximum Mean Discrepancy (MMD)-critic [223] and Pertinent Negatives [224]. C-CHVAE and DiCE perturb the input features to identify faithful counterfactuals, meaning that they do not represent local outliers and are "connected to regions with substantial data density", and are diverse enough to approximate local decision boundaries. Evasion-Prone Samples Selection detects the instances close to the classification boundaries that can be easily misclassified if slightly perturbed. MMD-critic utilises the maximum mean discrepancy and an associated witness function to identify the portions of the input space most misrepresented by the underlying model. Pertinent Negatives highlights what should be minimally and necessarily absent to justify the classification of an instance. For example, the absence of glasses is a necessary condition to say whether a person has good sight. The input data are modified by removing some parts. The pertinent negatives are those perturbations that enhance the prediction accuracy.

Finally, some XAI methods produce mixed explanations by approximating a black-box model with simpler, more comprehensible models that the end-users can inspect to assess the contribution of each feature. Local Interpretable Model-Agnostic Explanations (LIME) [22] explains the prediction of any classifier by learning a local self-interpretable model (such as linear models or decision trees), referred to as white-box models, trained on a new dataset that contains interpretable representations of the original data. These representations can be the binary vectors highlighting the presence or absence of certain characteristics, such as words in texts or super-pixels (contiguous patch of similar pixels) in images. The black-box model can be explained through the weights of the white-box estimator that does not need to work globally, but it should approximate the black-box well in the vicinity of a single instance. However, the authors proposed the Sub-Modular Pick (SP-LIME) to select, from an original dataset, a representative non-redundant explanation set of instances that is a global representation of the model.

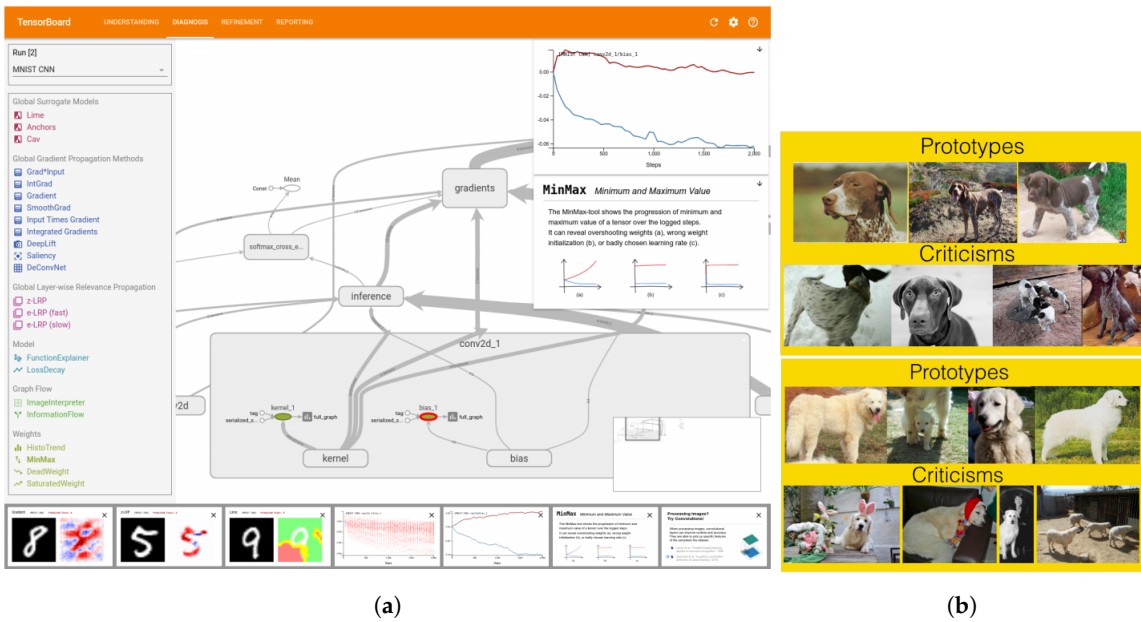

(a)            (b)

**Figure 14.** Examples of mixed explanations generated by model-agnostic XAI methods which consists of a combination of visual and textual explanations in (**a**) Rivelo [215] interactive interfaces; or (**b**) MMD-critic [223] a selection of prototypes from the input data.

**Table 17.** Post hoc model-agnostic XAI methods generating mixed explanations, classified according to the type of problem (C: classification; R: regression), scope (G: global; L: local) and input data (NC: numerical/categorical; P: pictorials; T: textual; TS: time series) of the underlying model.

| Method for Explainability | Authors | Ref | Year | Scope | Problem | Input |
|---|---|---|---|---|---|---|
| Bayesian Teaching | Yang and Shafto | [216] | 2017 | G | C | NC |
| C-CHVAE | Pawelczyk et al. | [220] | 2020 | L | C | NC |
| Combinatorial Methods | Kuhn et al. | [212] | 2020 | L | C/R | NC |
| DiCE | Mothilal et al. | [221] | 2020 | L | C | NC |
| Evasion-Prone Samples Selection | Liu et al. | [222] | 2018 | G | C | T |
| ExplAIner | Spinner et al. | [214] | 2019 | G | C/R | P; NC; TS |
| Functional ANOVA Decomposition and Variable Interaction Network Graph | Hooker | [211] | 2004 | G | C/R | NC |
| Justification Narratives | Biran and McKeown | [213] | 2014 | G | C | NC |
| LIME | Ribeiro et al. | [22] | 2016 | L | C | P; T |
| MMD-Critic | Kim et al. | [223] | 2016 | L | C | P |
| Neighbourhood-Based Explanations | Caruana et al. | [219] | 1999 | L | C | NC |
| Pertinent Negatives | Dhurandhar et al. | [224] | 2018 | L | C | P; NC |
| Rivelo | Tamagnini et al. | [215] | 2017 | L | C | T |
| SBQ | Khanna et al. | [217] | 2019 | L | C | P; NC |
| SCO | Bien et al. | [218] | 2011 | L | C | P; NC |

### 7.2. Model-Specific XAI Methods Based on Neural Networks

Some XAI methods have attempted at explaining the inferential process of neural networks with mixed explanations (see Figure 15 and Table 18). Attention Alignment [2] produces explanations in the form of attention maps, highlighting the parts of a scene that matter to a control DNN utilised in self-driving cars in combination with a perception DNN, and textual explanations such as "The car heads down the street because it is clear". The perception DNN combines the data received from cameras and other sensors, such as radars and infrared, to "understand" the environment and generate manoeuvring commands. The control DNN is trained to identify the presence of specific objects, such as road signs, and obstacles like pedestrians and bikers, that influence the output of the perception network. Similarly, the Pointing and Justification Model (PJ-X) [225] and Image Caption Generation with Attention Mechanism [35], both designed for VQA tasks, provide joint textual rationale generation and attention-map visualisation. The attention-maps are extracted from a CNN, which performs the object recognition in images, whereas the

textual justifications are produced by an LSTM network as image captions. Refs. [226,227] proposed two methods for replacing a DNN with a deterministic finite automaton that can be visualised as a graph where each node represents a cluster of values in the output space and the edges represent the presence of shared patterns in a network's internal layers between these clusters. Another method justifies the prediction of a new instance by identifying the three most similar training samples based on the cosine distance of the activation values of hidden neurons related to the training data [228]. Lastly, Representer Points [229] selects a set of prototypes from a training set by calculating linear combinations of the network's activation values and by choosing the training instances with the large values. The weights of the linear combinations are set to capture the importance of each training instance on the learned parameters of the network.

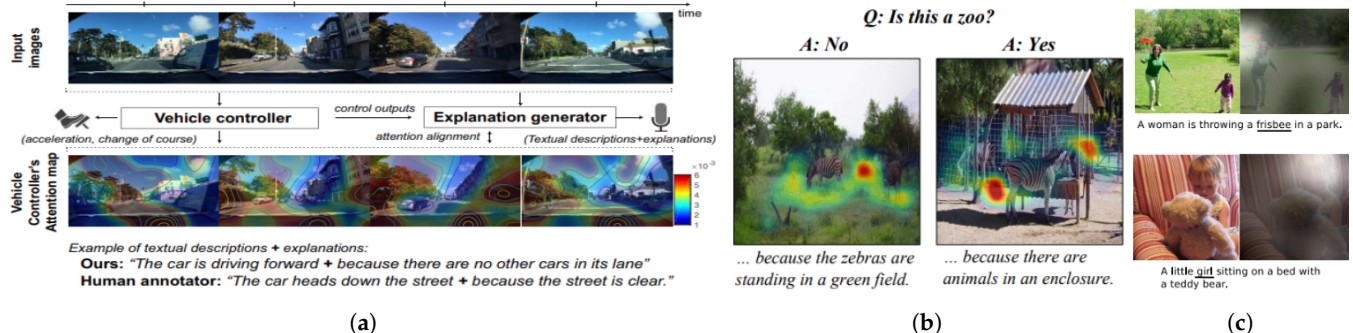

(**a**)          (**b**)          (**c**)

**Figure 15.** Examples of mixed explanations, consisting of combinations of heatmaps and textual captions, generated by XAI methods for neural networks, which highlight the most relevant parts of the input images. (**a**) Attention Alignment [2]; (**b**) PJ-X [225]; (**c**) Attention Mechanism [35].

**Table 18.** XAI methods for neural networks generating mixed explanations, classified according to the stage (AH: ante hoc; PH: post hoc), type of problems (C: classification; R: regression), scope (G: global; L: local) and input data (NC: numerical/categorical; P: pictorials; T: textual; TS: time series) of the underlying model.

| Method for Explainability | Authors | Ref | Year | Stage | Scope | Problem | Input |
|---|---|---|---|---|---|---|---|
| Activation Values of Hidden Neurons | Tamajka et al. | [228] | 2019 | AH | L | C | P |
| Attention Alignment | Kim et al. | [2] | 2018 | PH | L | C | P |
| Deterministic Finite Automaton | Mayr and Yovine | [226] | 2018 | PH | L | C | NC |
| DFAs | Omlin and Giles | [227] | 1996 | PH | G | C | NC |
| Image Caption Generation w/ Attention Mechanism | Xu et al. | [35] | 2015 | PH | L | C | P |
| PJ-X | Park et al. | [225] | 2018 | AH | L | C | P |
| Representer Points | Yeh et al. | [229] | 2018 | PH | L | C | P |

### 7.3. Other Model-Specific XAI Methods

Some scientific studies are devoted to develop XAI methods producing mixed-explanations for models based on learning algorithms other than neural networks (Table 19).

#### 7.3.1. Rule-Based System

ExpliClas [230] is a visual interface designed to explain, in an instance-based manner, rule-based classifiers (such as those algorithms extracting DTs from data, such as C4.5 or CART). The rules are shown as DTs and a natural language generator returns textual explanations of the fired rules. Exception-Directed Acyclic Graphs (EDAGs) [231] is an empirical induction tool that generates rules from the knowledge base of expert systems to create comprehensible knowledge structures in the form of graphs. The nodes are premises, some of which have attached conclusions, leaves are conclusions, and edges represent exceptions to some node. The "meaning" of each node can be easily determined by following its path back to the root and by inspecting its child nodes, whilst the rest of the graph is irrelevant.

### 7.3.2. Ensembles

Tree Space Prototypes (TSP) [232] selects prototypes from a training dataset to explain the prediction made by ensembles of DTs and gradient boosted tree models on a new observation. The authors proposed a metric to quantify the contribution of the predictions made by each DT to the ensemble's prediction. The metric is based on the weighted average of the number of trees in the ensemble, assigning the new observation to the same output class. By following the path root-to-leaf of the most relevant DT, it is possible to determine the values of the features deemed relevant by the tree for predicting the class of the new instance and select a prototype having the same values.

### 7.3.3. Support Vector Machines

SVM+Prototypes [233] uses a clustering algorithm to detect the prototype vectors for each class. These vectors are combined with the support vectors using geometric methods to define ellipsoids in the input space. The ellipsoids are transformed to IF-THEN rules as their defining mathematical equations, so a rule looks like 'If $AX_1^2 + BX_2^2 + CX_1X_2 + DX_1 + EX_2 + F \leq G$ Then $Class_1$'.

### 7.3.4. Bayesian and Hierarchical Networks

Probabilistically Supported Arguments (PSA) [234] is based on a two-phase algorithm for extracting probabilistically explanatory supported arguments from a Bayesian network. In the first phase, a support graph is constructed from the network for a particular variable of interest. In the second phase, given a set of observations, arguments are built from that support graph. To do so, the algorithm defines a logical language and a set of rules built from the support graph by following its edges and nodes. The parents of a node are the rule conditions, the node itself is the rule's outcome. Only the parents supported by pieces of evidence are considered. Then, an ASPIC+ framework for structured argumentation is instantiated. Arguments can attack each other on the conclusion variable and defeat can be based on the inferential strength of the arguments which can be computed with two types of measures: "incremental measures" which assign a number to the weight of the evidence (the likelihood ratio is an example of these measures) and "absolute measures" which assign strength based on posterior probability, such as the posterior for instance. Such arguments can help interpret and explain the relationship between hypotheses and evidence modelled in the Bayesian network.

**Table 19.** XAI methods for data-driven approaches generating mixed explanations, classified according to the construction approach (learning algorithm), stage (AH: ante hoc; PH: post hoc), type of problem (C: classification; R: regression), scope (G: global; L: local) and input data (NC: numerical/categorical; P: pictorials; T: textual; TS: time series).

| Method for Explainability | Authors | Ref | Year | Construction Approach | Stage | Scope | Problem | Input |
|---|---|---|---|---|---|---|---|---|
| EDAGs | Gaines | [231] | 1996 | Rule-based system | AH | G | C | NC |
| ExpliClas | Alonso | [230] | 2019 | Rule-based system | PH | L | C | NC |
| Probabilistically Supported Arguments | Timmer et al. | [234] | 2017 | Bayesian networks | PH | G | C | NC |
| SVM+Prototypes | Núñez et al. | [233] | 2002 | SVM | PH | G | C | NC |
| Tree Space Prototypes | Tan et al. | [232] | 2016 | Ensembles | PH | L | C | NC |

### 7.4. Self-Explainable and Interpretable Methods

A few ante hoc XAI methods modify the structure of ML models that generate mixed outputs (Table 20). Bayesian Case Model (BCM) [235] is a method for explainability for Bayesian case-based reasoning, prototype classification and clustering. BCM learns prototypes, corresponding to the observations that best represent clusters in a dataset, by performing joint inference on cluster labels, prototypes and important features. Generalised Additive Models [17] and their extension with pairwise interactions (GA²Ms) [18,236] are linear combinations of simple models, called "shape functions", trained on a single feature (GAMs) or up to two features (GA²Ms). Their simple structure allows the end-user to easily understand the contribution of individual features to the predictions and to visualise

them, together with the shape functions, with bar- and line-charts. Multi-Run Subtree Encapsulation, which comes from the genetic programming (GP) realm, was proposed in [237] as a way to generate simpler tree-based GP programs. If the tree contains sub-trees of different makeup but evaluating the same vector of results, they are to be considered as the same sub-tree. This reduces, according to the authors, the complexity of the entire tree structure and the resulting expressions, in favour of explainability. Mind the Gap Model (MGM) [238] is a method for interpretable feature extraction and selection. The goal is to split the observation into clusters while returning the list of dimensions that are important for distinguishing them. The results are presented as a mix of numbers, which are the relevance values of each dimension, texts and graphs that represent the dimensions themselves. For example, in a classification problem of images representing the four seasons, MGM returns samples of images belonging to each class (spring, summer, autumn and winter) together with the list of their relevant features (such as snow, sun and flowers) and the relevance values of each feature per target class (snow has a high relevance value for the class "winter"). A hybrid DL approach [239] uses a model to automatically identify meaningful, hand-crafted, high-level symbolic features of the input dataset. These features are subsequently employed by a more interpretable learning model.

**Table 20.** Ante hoc XAI methods generating white-box models generating mixed explanations, classified according to the type of problem (C: classification; R: regression), scope (G: global; L: local) and input data (NC: numerical/categorical; P: pictorials; T: textual; TS: time series) of the underlying model.

| Method for Explainability | Authors | Ref | Year | Scope | Problem | Input |
|---|---|---|---|---|---|---|
| BCM | Kim et al. | [235] | 2014 | G | C | P; T |
| EDAGs | Gaines | [231] | 1996 | G | C | NC |
| GAMs | Lou et al., Lou et al. | [17,18] | 2012 | G | C/R | NC |
| GA$^2$Ms | Lou et al., Caruana et al. | [236] | 2015 | G | C/R | NC |
| Hybrid Deep Learning | Campagner and Cabitza | [239] | 2020 | G | C/R | NC |
| MGM | Kim et al. | [238] | 2015 | G | C | P; NC; T |
| Multi-Run Subtree Encapsulation | Howard and Edwards | [237] | 2018 | G | C | NC |

## 8. Final Remarks and Recommendations

The term XAI groups together the scientific body of knowledge developed while searching for methods to explain the inner logic of either a learning algorithm, a model induced from it, or a knowledge-based approach for inference and it is now generally recognised as a core area of AI. Several studies are published every year, with many workshops and conferences organised around the world to present novel methods and disseminate findings. This has led to the production of an abundance of XAI methods. Scholars have attempted to comprehensively organised them, however, all these classification systems lack an important discriminative dimension, which is the output format of the explanations generated by these methods. This review attempted to fill this gap by organising them according to this dimension in addition to the traditional dimensions, such as scope and stage, within a hierarchical system. Since the early 1980s and 1990s, with research only concerned with textual explanations, to nowadays, scholars have been targeting new explanation formats whose strengths and weaknesses are summarised in Figure 16.

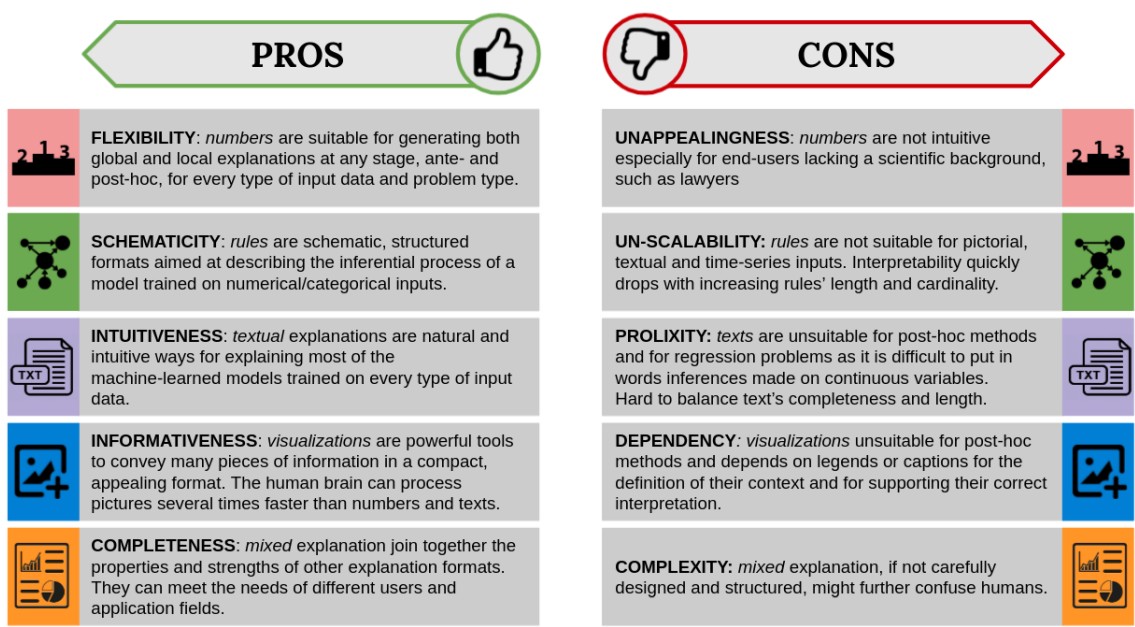

**Figure 16.** Summary of the pros and cons associated to each explanation format, namely numeric, rules, textual, visual and mixed explanations.

For each of these formats, scholars designed, deployed and tested several solutions, such as saliency masks, attention maps, heat-maps, feature maps, as well as graphs, rules sets, trees and dialogues. These advances were aimed at meeting the needs of different types of end-users operating in various fields, such as lay people, doctors and lawyers, and adapting explanations to their domains of application. Despite these improvements, there are still important gaps that must be addressed. There is not a consensus among scholars on what an explanation exactly is and which are the salient properties that must be considered to make it understandable for every end-user. Therefore, the search for an effective way to learn representations of the inferential process of data-driven models is still ongoing [240]. This holds for every explanation format, either textual, visual, numerical or rule-based, particularly for those that are generated after a model has been induced by employing deep-learning neural networks. In accordance with [1], we believe that scholars have produced enough material and knowledge to construct a generally applicable framework for XAI to guide the development of end-to-end XAI methods flexible enough to adapt to various contexts, fields of application and type of end-users, rather than keep creating isolated methods that remain only fragments of a broad solution, as shown in Figure 17. The world is complex, and thus so must be an XAI framework with a universal outlook. Distinct users look for explanations that achieve different purposes. Domain experts, such as doctors and bankers, require access to information that varies from that sought by decision makers, such as managers and regulators. Consumers do not have any control over the models but are affected by their inferences, so they need to trust them. Other users, such as the AI practitioners or managers, might be in charge of auditing and certifying that a model complies with regulations and quality standards. Hence, they must know its inner functioning. Researchers instead use the models to discover novel knowledge. The field of the application adds further complexity as it involves the search for solutions to problems that vary considerably, even within the same field, and are experienced by operators with distinct characteristics and needs, ranging from governments to business enterprises and academia. These organisations operate on different domains of knowledge and data that vary in type and quality, such as images, tables, text documents or online posts. Furthermore, the culture and structure of the organisations adopting AI-powered technologies must be considered to make the AI–human integration process as smooth as possible. It is necessary to analyse all these factors in-depth to identify the explanation requirements for each of them. These requirements include but are not limited to the five

dimensions of the XAI methods discussed in this literature review, namely the stage, scope, problem, input data and format of the explanation. There are several other factors relative to the quality and effectiveness of an explanation itself that cannot be ignored [8]. Once this analysis is finished, it will be possible to determine the best explanation format for each situation and provide a general solution to the quest for explainability in the AI field.

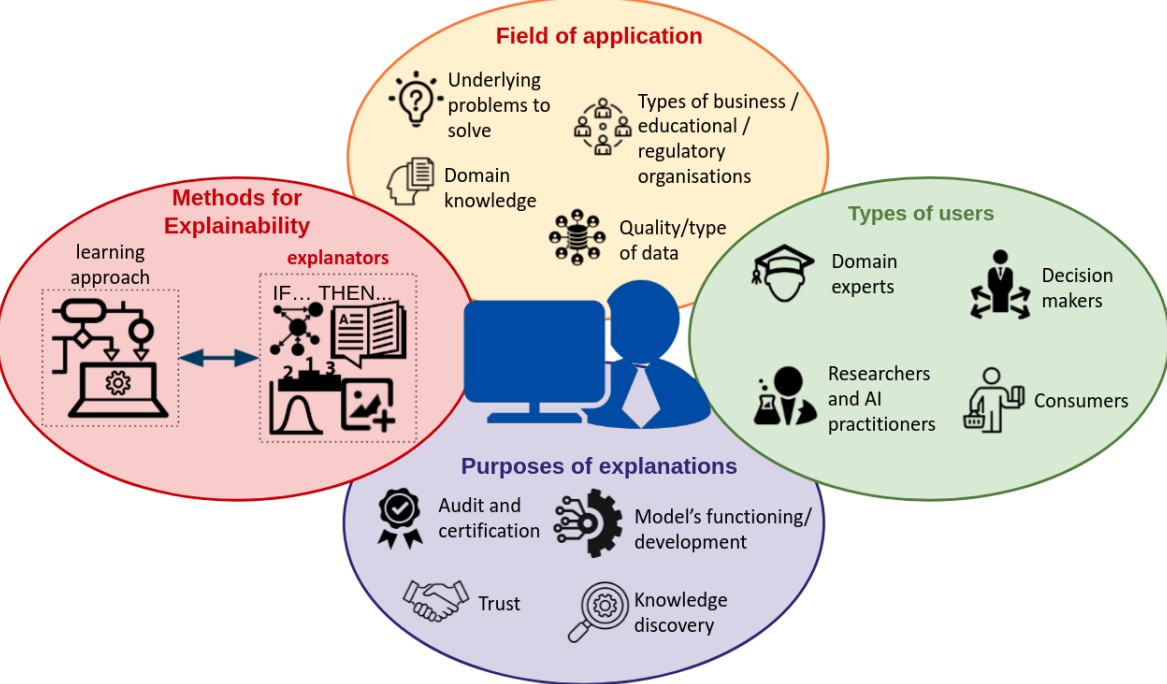

**Figure 17.** Diagram of the factors affecting the selection of XAI methods.

**Author Contributions:** Conceptualisation, G.V. and L.L.; methodology, G.V.; investigation, G.V.; writing—original draft preparation, G.V.; writing—review and editing, L.L.; visualisation, G.V.; supervision, L.L. All authors have read and agreed to the published version of the manuscript.

**Funding:** This research received no external funding.

**Institutional Review Board Statement:** Not applicable.

**Informed Consent Statement:** Not applicable.

**Data Availability Statement:** Data are contained within the article.

**Conflicts of Interest:** The authors declare no conflict of interest.

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
