# Peer review of "Classification of Explainable Artificial Intelligence Methods through Their Output Formats"

_make, doi:10.3390/make3030032_

Round 1

Reviewer 1 Report

Summary:

This manuscript is a review of research area for classification of explainable AI (XAI) methods focusing on the output formats. Authors investigated the literatures from Google Scholar with respect to the categories regarding the outputs from DNN or CNN. They categorized the output into numerical, rule-based, textual visual and mixed ones. Each categories were explained in the manuscript with references from the literatures. The authors finally insisted that the researcher in the XAI area should consider the output format to unify the algorithms and strategies invented in the past research.

Comments:

  • This manuscript reviewed the perspective of XAI researches well. The overviews of XAI algorithms picked up in the manuscript are helpful for the readers. I agreed the conclusions of the authors. However, it would be better to add some more background of XAI regarding the application examples of the different output categories right after the introduction. It was not easy to image the final goal of the review report, that is, the different categories from the explanations of algorithms. A suggestion is to show typical applications from literatures and explain the difficulties to unify the output formats into a common one. Then, we can clearly understand the main focus point of the investigations in the manuscript.
  •  In the abstract, "Explainable". In the manuscript, "eXplainable". Please use one of those.
  • In page 2, "tutorials on XAI were discarded.". It would be better to refer some literatures in the tutorial part.
  • Please check English readability again. For example,  the part "... into two sets the activation values ..." in page 6, line 207 is not understandable.

Author Response

Please find my answers in red.

  • This manuscript reviewed the perspective of XAI researches well. The overviews of XAI algorithms picked up in the manuscript are helpful for the readers. I agreed the conclusions of the authors. However, it would be better to add some more background of XAI regarding the application examples of the different output categories right after the introduction. It was not easy to image the final goal of the review report, that is, the different categories from the explanations of algorithms. A suggestion is to show typical applications from literatures and explain the difficulties to unify the output formats into a common one. Then, we can clearly understand the main focus point of the investigations in the manuscript. ANSWER: I have added lines 57-68 on page 2.
  •  In the abstract, "Explainable". In the manuscript, "eXplainable". Please use one of those.  ANSWER: Well spotted, I’ve fixed it.
  • In page 2, "tutorials on XAI were discarded.". It would be better to refer some literatures in the tutorial part. ANSWER: I have added a few references at line 90 on page 3.
  • Please check English readability again. For example,  the part "... into two sets the activation values ..." in page 6, line 207 is not understandable. ANSWER: Thank you for pointing this out, I’ve fixed this statement and revised the entire manuscript.

Reviewer 2 Report

1. Authors undertook a very ambitious task, ie "Scholars have attempted to organized them in a comprehensive manner, but all these classification systems lack an important discriminative dimension, which is the output format of the explanations generated by these methods. This review attempted to fill this gap by organizing them according to this dimension beside the traditional dimensions, such as scope and stage, within a hierarchical system ”.
2. Unfortunately, it was not an easy task, the effects of which are also not visible. The presented manuscript did not live up to this purpose.
3. The article is too long. There are frequent repetitions, inaccuracies. The article still requires a lot of work and a thorough arrangement.
4. Many of the authors' announcements are not extended further in the work, eg "This work concludes by critically identifying limitations of various XAI methods, by providing recommendations and possible future research directions." - in my opinion, it does not end like this!
5. The abstract lacks the method used in the selection of research papers. There is also a lack of a realistic goal of the article and conclusions that sum up the research work.
6. The research gap should be described at the end of presenting the background for the research problem. Most often it is an introduction, but not a conclusion.
7. The hierarchical system described by the authors is easiest to present in the form of a graphic diagram. This is missing.
8. Conclusions should be supported by own (not someone else's) research. Otherwise, they may suggest that it is a discussion and not a conclusion.
9. It is not known to which research results the presented conclusions relate. 

Author Response

Please find my answers in red.

  1. Authors undertook a very ambitious task, ie "Scholars have attempted to organized them in a comprehensive manner, but all these classification systems lack an important discriminative dimension, which is the output format of the explanations generated by these methods. This review attempted to fill this gap by organizing them according to this dimension beside the traditional dimensions, such as scope and stage, within a hierarchical system ”. ANSWER: Ok, thank you for this comment.
  2. Unfortunately, it was not an easy task, the effects of which are also not visible. The presented manuscript did not live up to this purpose. ANSWER: Thank you for this comment. We made a few changes, as outlined in the answers to the next comments, to improve the manuscript. We hope that now this issue is fixed.
  3. The article is too long. There are frequent repetitions, inaccuracies. The article still requires a lot of work and a thorough arrangement. Thank you for pointing this out, I’ve revised the entire manuscript. ANSWER: We understand that the length of the manuscript can be overwhelming, but the XAI scientific literature is already extensive and any general systematic review must deal with hundreds of studies, so cannot be short.
  4. Many of the authors' announcements are not extended further in the work, eg "This work concludes by critically identifying limitations of various XAI methods, by providing recommendations and possible future research directions." - in my opinion, it does not end like this! ANSWER: We have completely revised the conclusion section and added two diagrams that should highlight its main points. We hope this helps in fixing this issue.
  5. The abstract lacks the method used in the selection of research papers. There is also a lack of a realistic goal of the article and conclusions that sum up the research work. The abstract has been updated as requested. ANSWER: The abstract has been revised as suggested.
  6. The research gap should be described at the end of presenting the background for the research problem. Most often it is an introduction, but not a conclusion. ANSWER: Please see lines 45-51 on page 2
  1. The hierarchical system described by the authors is easiest to present in the form of a graphic diagram. This is missing. ANSWER: Good point, I’ve added Figure 2 on page 4.
  2. Conclusions should be supported by own (not someone else's) research. Otherwise, they may suggest that it is a discussion and not a conclusion. ANSWER: Please check the new conclusion section. We hope that the additions will answer this and the next comment. 
  3. It is not known to which research results the presented conclusions relate. 

Reviewer 3 Report

The manuscript described a collection of explainable artificial intelligence methods together with referred illustrations adequately.  The review is written in a clear and didactic manner and brings together several interesting avenues of work for future readers of this paper.  Overall, it scientifically sounds well.  It is an informative review that deserves publication if the following comments have been addressed.

  1. There is a collection of XAI methods described in the manuscript. Authors have introduced them in terms of their methodology.  It will be more informative if authors can organize those methods as a tree together with Figure 1.  Thus, readers will be able to have an overall picture before they look into the details of the paper.
  2. It is great that authors have provided a number of figures to illustrate the referred XAI methods. However, the text in those figures are likely too small or too implicit to give a straightforward explanation to readers.  Also, many of their legends are too short or simple.  Readers cannot fully understand the content and the abbreviations of the figure without having to refer to the main text.
  3. It is recommended to provide figure number and code in the text (e.g. Figure 4(b)), so that readers can directly refer to the sub-figure when they are reading at some point (e.g. session 3.1).
  4. Authors put the Random Forest method in the session of ensembles which is no doubtable. It is however also reasonable to put it in the session of Rule-based explanations.

Author Response

Please check the answers in red.

  1. There is a collection of XAI methods described in the manuscript. Authors have introduced them in terms of their methodology.  It will be more informative if authors can organize those methods as a tree together with Figure 1.  Thus, readers will be able to have an overall picture before they look into the details of the paper. ANSWER: Good point, I’ve added Figure 2 on page 4. 
  2. It is great that authors have provided a number of figures to illustrate the referred XAI methods. However, the text in those figures are likely too small or too implicit to give a straightforward explanation to readers.  Also, many of their legends are too short or simple.  Readers cannot fully understand the content and the abbreviations of the figure without having to refer to the main text. ANSWER: I’ve increased the size of all the figures and revised the captions. I hope that this is ok.
  3. It is recommended to provide figure number and code in the text (e.g. Figure 4(b)), so that readers can directly refer to the sub-figure when they are reading at some point (e.g. session 3.1). ANSWER: That has been fixed, thank you for highlighting it.
  4. Authors put the Random Forest method in the session of ensembles which is no doubtable. It is however also reasonable to put it in the session of Rule-based explanations. ANSWER: Some XAI methods on random forests are not classified under the rule-based explanations because the explanations that they provide are not based on rules, but are numeric, visual, textual or a mix of them. For instance, the method called Feature Tweaking explains the logic of a model based on the random forest learning algorithm by returning as an explanation the ‘tweaking cost’ of each instance, which consists of the size of a linear shift that must be applied to the instance to be classified under another class. Thus, this ‘tweaking cost’ is a number and the method is classified under numeric explanations for ensemble algorithms. Moving it under rule-based explanations would be an error. I’ve added this clarification in the manuscript at lines 173-186 on page 5.

Reviewer 4 Report

The authors of the paper have done extensive and quality work by giving a review of research in the field of Explainable Artificial Intelligence. They developed a new and very interesting taxonomy of presenting XAI methods, taking into account different input data characteristics and specific ways of presenting explanations of machine learning models, using representative state of the art illustrations. The work is quite extensive, but it certainly provides interesting information to researchers presented in an original way.

Author Response

ANSWER: Thank you very much for your positive feedback, we really appreciate it. We are aware of the size of the paper and we hope that this will not represent a limit and the other researchers will find it interesting, as you did. However, we really want to present a comprehensive overview of the state of the art in the XAI field and, given the sheer number of studies published so far, we could not make this manuscript any shorter.

Round 2

Reviewer 1 Report

The authors revised the manuscript appropriately by responding my comments and questions. I think it is ready for publication.

Reviewer 2 Report

The Authors made changes. I recommend the article for publication.